# A model-based factorization method for scRNA data unveils bifurcating transcriptional modules underlying cell fate determination

Jun Ren[1,2,3], Ying Zhou[1,2], Yudi Hu[1], Jing Yang[1], Hongkun Fang[4], Xuejing Lyu[1], Jintao Guo[4], Xiaodong Shi[3], Qiyuan Li[1,2]*

[1]National Institute for Data Science in Health and Medicine, School of Medicine, Xiamen University, Xiamen, China; [2]Department of Hematology, The First Affiliated Hospital of Xiamen University and Institute of Hematology, School of Medicine, Xiamen University, Xiamen, China; [3]School of Informatics, Xiamen University, Xiamen, Xiamen, China; [4]Department of Scientific Research Management, Weifang People's Hospital, Shandong Second Medical University, Weifang, China

## eLife Assessment

MGPfact[XMBD] is a novel computational method for investigating cell evolutionary trajectory for scRNA-seq samples. It is **important**, with several potential future applications. The authors benchmarked this method using synthetic and real-world samples and showed superior performance for some of the tasks in cell trajectory analysis compared to other methods with **compelling** evidence.

*For correspondence:
qiyuan.li@xmu.edu.cn

**Abstract** Manifold-learning is particularly useful to resolve the complex cellular state space from single-cell RNA sequences. While current manifold-learning methods provide insights into cell fate by inferring graph-based trajectory at cell level, challenges remain to retrieve interpretable biology underlying the diverse cellular states. Here, we described MGPfact[XMBD], a model-based manifold-learning framework and capable to factorize complex development trajectories into independent bifurcation processes of gene sets, and thus enables trajectory inference based on relevant features. MGPfact[XMBD] offers a more nuanced understanding of the biological processes underlying cellular trajectories with potential determinants. When bench-tested across 239 datasets, MGPfact[XMBD] showed advantages in major quantity-control metrics, such as branch division accuracy and trajectory topology, outperforming most established methods. In real datasets, MGPfact[XMBD] recovered the critical pathways and cell types in microglia development with experimentally valid regulons and markers. Furthermore, MGPfact[XMBD] discovered evolutionary trajectories of tumor-associated CD8[+] T cells and yielded new subtypes of CD8[+] T cells with gene expression signatures significantly predictive of the responses to immune checkpoint inhibitor in independent cohorts. In summary, MGPfact[XMBD] offers a manifold-learning framework in scRNA-seq data which enables feature selection for specific biological processes and contributing to advance our understanding of biological determination of cell fate.

## Introduction

Data-mining of single-cell RNA sequencing (scRNA-seq) is often transformed into learning of lower-dimensional embedding (*Becht et al., 2019*; *Haghverdi et al., 2015*; *Maaten and Hinton, 2008*) of the expression vectors, which represents the variation in the cellular space and helps explain the biological background. Previous single-cell studies used various embedding methods to characterize and visualize clustering of cells with unique biological functions (*Saelens et al., 2019*). Among the existing methods, graph-based embedding can better capture nonlinear biological signals among cells hence yielded more insights of the diversity of cells. More recent studies also use graph-based embedding (*Costa et al., 2018*) to reveal the dependency among cells and thereby reconstruct the evolutionary trajectory in the cellular space, which helps in understanding the determination of cell fate in development, differentiation and cancer.

To date, more and more manifold-learning methods are developed to infer lower-dimensional graphic embedding (manifolds) of scRNA-seq data, and yielded a number of trajectories corresponding to important cellular processes, such as TSCAN (*Ji and Ji, 2016*), DPT (*Haghverdi et al., 2016*), and scShaper (*Smolander et al., 2022*) belong to linear topological classes and reveal major linear pathways based on embedding spaces or cell clustering. TSCAN employs the construction of minimum spanning trees to discover pathways, while DPT reconstructs cellular trajectories using random walks, and scShaper integrates multiple pseudo-temporal solutions to deduce the shortest trajectory within a linear context. Additionally, there have been many approaches capable of inferring complex tree topological structures, such as the widely used Monocle series of algorithms includes Monocle 2 and 3 (*Cao et al., 2019*; *Qiu et al., 2017b*, *Qiu et al., 2017a*). They leverage sophisticated graphing techniques to map intricate cell hierarchies; Monocle 2 creates DDRtree based on reversed graph embedding techniques, while Monocle 3 utilizes UMAP (*Becht et al., 2019*) for embedding. TinGa (*Todorov et al., 2020*) and scFates (*Faure et al., 2023*) represent more recent innovations. TinGa utilizes the Growing Neural Gas (GNG) algorithm (*Fritzke, 1994*) to construct an adaptive graph structure that effectively captures the density structure of the dataset. scFates, streamlines pseudotime analysis with flexible tree learning options, advanced feature extraction tasks, and specific functionalities for fork analysis.

Moreover, recent studies based on RNA velocity have provided insights into cell state transitions. These methods measure RNA synthesis and degradation rates based on the abundance of spliced and unspliced mRNA, such as CellRank (*Lange et al., 2022*). Nevertheless, current RNA velocity analyses are still unable to resolve cell-fates with complex branching trajectory. Deep learning methods such as scTour (*Li, 2023*) and TIGON (*Sha et al., 2024*) circumvent some of these limitations, offering continuous state assumptions or requiring prior cell sampling information.

Despite these advances, trajectory prediction remains a major challenge in single-cell analysis. First, graph-based trajectories represent synergic effects of multiple biological processes, making it difficult to disentangle the effects of specific process, hence limited model interpretability and the power to gain novel biological insights. Second, the inference of trajectory is highly dependent on the biases in the gene-selection, whereas conventional statistical feature-selection methods are less efficient for the learning of complex topologies, and adds to the difficulty of suggesting candidate genes for downstream functional study. Additionally, many approaches require additional prior information, which further limits the applicability.

Here, we describe MGPfact[XMBD] (Factorization based on Mixtures of Gaussian Processes), a model-based, unsupervised manifold-learning method which factorizes complex cellular trajectories into interpretable bifurcation Gaussian processes of transcription, and thereby enables discovery of specific biological determinants of cell fate. In the validation datasets, MGPfact recapitulated developmental trajectory of microglia and recovered key regulatory factors which have been proved experimentally. Moreover, MGPfact discovered highly specific subtypes of tumor-associated CD8[+] T cells which are associated with benefit to cancer immunotherapy.

Bring together, MGPfact is a knowledge discovery tool which conducts manifold-learning and factorization simultaneously. MGPfact offers two advantages in future scRNA-seq analyses: first, it provides highly interpretable, factorizable cellular trajectories with matched gene expression modules; then, it provides efficient feature-selection for graph-based embedding, thus enhancing our understanding of the determination of cell fate.

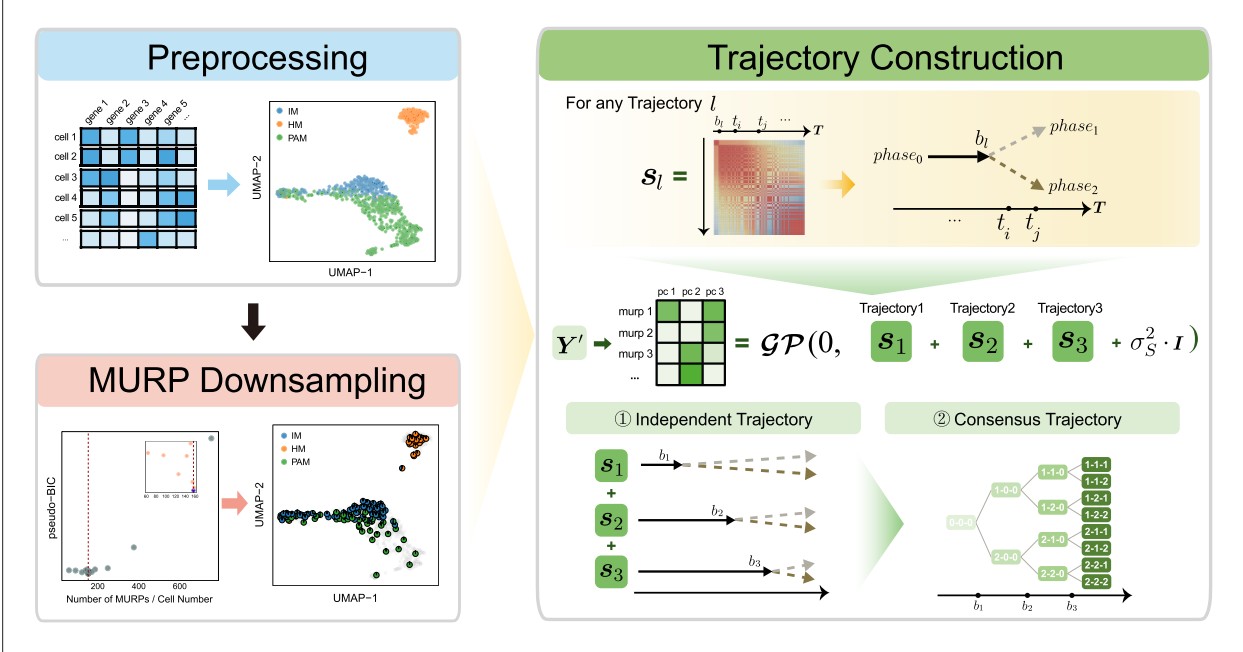

**Figure 1.** Overview of MGPfact workflow. The complete workflow comprises two major stages: minimum unbiased representative points (MURP) downsampling with preprocessed data and trajectory reconstruction. In the stage of trajectory reconstruction, the single-cell RNA sequencing (scRNA-seq) data were first factorized into independent bifurcation processes based on mixtures of Gaussian processes, which were then merged into a consensus trajectory.

## Results

### Design of MGPfact

The analytical pipeline of MGPfact consists of two major stages (*Figure 1*). An algorithmic description is given in Algorithm 1.

First, we performed downsampling of the preprocessed scRNA-seq data $\boldsymbol{Y}$ to yield a $M$-by-$N$ expression matrix $\boldsymbol{Y}'$ based on the 'minimum unbiased representative points (MURP)' as described previously (*Ren et al., 2022*), where $M$ representative points were considered as landmarks of the cellular trajectory and $N$ is the number of genes. Then, we computed $L$ principal components (PCs) of the downsampled expression matrix to obtain the matrix $\boldsymbol{Y}^* = \{\boldsymbol{y}_1^*, \boldsymbol{y}_2^*, \boldsymbol{y}_3^*, \ldots \boldsymbol{y}_L^*\}$ ($M$-by-$L$),

$$\boldsymbol{y}_l^* = \boldsymbol{Y}' \cdot \boldsymbol{v}_l \tag{1}$$

where $\boldsymbol{v}_l$ is projection vector, $\boldsymbol{y}_l^*$ serve as the $l$-th initial state of embedding.

Next, we used typical Gaussian Process Regression of $\boldsymbol{y}_l^*$ on pseudotime $\boldsymbol{T}$:

$$\boldsymbol{y}_l^* = f(\boldsymbol{T}) + \varepsilon \tag{2}$$

where $f(\boldsymbol{T})$ is a Gaussian Process (GP) with covariance matrix $\boldsymbol{S}$.

$$f(\boldsymbol{T}) = \mathcal{GP}\left(0, \boldsymbol{S} + \sigma_S^2 \cdot \boldsymbol{I}\right) \tag{3}$$

And for all features:

$$p\left(\boldsymbol{Y}^*, f(\boldsymbol{T})\right) = p\left(\boldsymbol{Y}^* \mid f(\boldsymbol{T})\right) \cdot p\left(f(\boldsymbol{T})\right) \tag{4}$$

where $p\left(\boldsymbol{Y}^* \mid f(\boldsymbol{T})\right)$ is defined as follows:

$$p\left(\boldsymbol{Y}^* \mid f(\boldsymbol{T})\right) = \prod_{l=1}^{L} p\left(\boldsymbol{y}_l^* \mid f(\boldsymbol{T})\right) = \prod_{l=1}^{L} \mathcal{N}\left(\boldsymbol{y}_l^* \mid 0, \boldsymbol{S} + \sigma_S^2 \cdot \boldsymbol{I}\right) \tag{5}$$

Specifically, we consider $S$ as a mixture of $L$ independent bifurcation Gaussian processes (**Schulz et al., 2018**),

$$S = \sum_{l=1}^{L} s_l \tag{6}$$

To cope with the bifurcation processes in cell fate, each of the Gaussian processes is defined with a bifurcation point at $b_l$, branching labels $c_l$, and the necessary hyperparameters. The branching labels $c_l \in \{0, 1, 2\}$, correspond to different phases and states of cell fate, where $c_l = 0$ corresponds the phase before branching, and $c_l \in \{1, 2\}$. corresponds to the two cellular states of the bifurcation process, respectively. For any landmark (MURP) $x$,

$$\begin{cases} c_{l,x} = 0, & t_x < b_l \\ c_{l,x} \in \{1, 2\}, & t_x \geq b_l \end{cases} \tag{7}$$

The covariance matrix $s_l$ for trajectory $l$ can be expressed as follows,

$$[s_l]_{x,y} = \mathcal{K}(t_x, t_y) \tag{8}$$

where $\mathcal{K}$ is a kernel function. We employ radial basis function (rbf) and polynomial kernel function (pl). We chose these two kernel functions for the effectiveness in handling nonlinear and polynomial relationships, achieving a balance between model performance and computational efficiency.

$$k_{rbf}(t_x, t_y) = \lambda_{rbf} \cdot e^{\left(-\alpha_{rbf}||t_x - t_y||^2\right)} \tag{9}$$

$$k_{pl}(t_x, t_y) = \left(\lambda_{pl} \cdot t_x^T t_y + c_{pl}\right)^{d_{pl}} \tag{10}$$

And the $\mathcal{K}(t_x, t_y)$ is calculated as follows:

$$\mathcal{K}(t_x, t_y) = \begin{cases} k_{rbf}(t_x, t_y) + k_{pl}(t_x, t_y) & t_x, t_y < b_l \\ k_{rbf}(t_x, t_y) + k_{pl}(t_x - b_l, t_y - b_l) & t_x, t_y > b_l, c_{l,x} = c_{l,y} \\ \dfrac{k_{rbf}(t_x, b_l) \cdot k_{rbf}(b_l, t_y)}{k_{rbf}(b_l, b_l)} & c_{l,x} \neq c_{l,y} \end{cases} \tag{11}$$

Therefore, $p(Y^* | f(T))$ is updated as follows:

$$p(Y^* | f(T)) = \prod_{l=1}^{L} \mathcal{N}\left(y_l^* | 0, \sum_{l=1}^{L} s_l^* + \sigma_S^2\right) \tag{12}$$

We infer all parameters by maximizing the posterior likelihood using Markov Chain Monte Carlo (MCMC) methods available in Mamba (**Brian, 2014**). The posterior distribution of pseudotime $T$ can be represented as:

$$p(T | Y^*) \propto p(Y^* | f(T)) \cdot p(f(T)) \tag{13}$$

where $p(Y^* | f(T))$ is the likelihood function of the observed data $Y^*$, and $p(f(T))$ is the prior distribution of the Gaussian process. This posterior distribution integrates the observed data with model priors, enabling inference of pseudotime and trajectory simultaneously. Due to the high autocorrelation of $T$ in the posterior distribution, we use Adaptive Metropolis within Gibbs (AMWG) sampling (**Roberts and Rosenthal, 2009**; **Tierney, 1994**). Other parameters are estimated using the more efficient SLICE sampling technique (**Neal, 2003**).

Algorithm 1 **MGPfact: infer cell fate trajectory**

INPUT: expression matrix $Y$, independent trajectories number $L$
OUTPUT: $\theta = \{T, B, C, \text{other hyperparameters}\}$ ; trajectory toplogy
1: initialize all parameters in $\theta$
2: covariance matrix $S$
3: object function $\mathcal{L} = 0$
4: $Y' \leftarrow$ MURP downsampling $Y_{p,n}$;
5: $Y^* \leftarrow$ PCA analysis
6: $\theta \leftarrow$ Opmized ObjectF
7: $Graph \quad \leftarrow$ Greate independent trajectory using $\theta$
8: **function** ObjectF $(Y^*, T, b_l, c_l)$
9: $\quad Q \leftarrow dim(Y^*)[2]$
10: $\quad$ **for** $q = 1; q < \mathcal{Q}; q++$ **do** $\qquad\qquad\quad \triangleright$ object function
11: $\quad\quad v \leftarrow [Y^*]_q$
12: $\quad\quad$ **for** $l = 1; l < L; l++$ **do**
13: $\quad\quad\quad s_l \leftarrow$ Cov$(T, b_\ell, c_\ell)$
14: $\quad\quad\quad S \leftarrow S + s_l$
15: $\quad\quad$ **end for**
16: $\quad\quad \mathcal{L} \leftarrow \mathcal{L} + MultivariateNormalPDF(v, 0, S)$
17: $\quad$ **end for**
18: $\quad$ return $\mathcal{L}$
19: **end function**
20: **function** Cov $(T, b_l, c_l)$ $\qquad\qquad\qquad \triangleright$ construct covariance matrix
21: $\quad P \leftarrow length(T)$
22: $\quad$ **for** $i = 1; i < P; i++$ **do**
23: $\quad\quad$ **for** $j = 1; j < P; j++$ **do**
24: $\quad\quad\quad s_l[i,j] \leftarrow KERNEL\left(t_i, t_j, b_l, c_l\right)$
25: $\quad\quad$ **end for**
26: $\quad$ **end for**
27: $\quad$ return $\quad s_l$
28: **end function**
29: **function** $KERNEL\left(t_i, t_j, b_l, c_{l,i}, c_{l,j}\right) \triangleright$ kernel function
30: $\quad$ **if** $t_i, t_j < b_l$ **then**
31: $\quad\quad$ return $\quad k_{rbf}(t_i, t_j) + k_{pl}(t_i, t_j)$
32: $\quad$ **else if** $\quad t_i, t_j > b_l \wedge c_{l,i} = c_{l,j}$ **then**
33: $\quad\quad$ return $k_{rbf}(t_i, t_j) + k_{pl}(t_i - b_l, t_j - b_l)$
34: $\quad$ **else if** $c_{l,i} \neq c_{l,j}$ **then**
35: $\quad\quad$ return $\frac{k_{rbf}(t_i, b_l) \cdot k_{rbf}(b_l, t_j)}{k_{rbf}(b_l, b_l)}$
36: $\quad$ **end if**
37: **end function**

Then, MGPfact can identify genes that have significant impacts on the branching events in the trajectories. We introduce a rotation matrix $R = \{r_1, r_2, ..., r_L\}$ to obtain factor score $w_l$ for each trajectory $l$ by rotating $Y^*$.

$$w_l = Y^* \cdot r_l + e_l^2 \tag{14}$$

For all trajectories,

$$p\left(W \mid Y^*\right) = \prod_{l=1}^{L}\left[\mathcal{N}\left(Y^* \cdot r_l + e_l \mid 0, s_l\right) \cdot \mathcal{N}\left(e_l \mid 0, \sigma_{error}^2\right)\right] \tag{15}$$

where $e_l$ is the error term for the $l$-th trajectory.

Specifically, the factor scores $w_l$ for each gene onto the $l$-th trajectory can be expressed using **equations (1) and (14)** as follows,

$$w_l = \left[Y' \cdot v_l\right] \cdot r_l + e_l^2 = Y' \cdot u_l + e_l^2 \tag{16}$$

Here, $u_l$ is used to represent the contribution (gene weight) of each gene to the $l$-th trajectory, thus enabling gene-selection based on the inferred trajectories.

Additionally, we can combine independent bifurcation processes to form a consensus diffusion tree to represent the trajectory of cell fate (**Supplementary file 1**).

## Performance evaluation of MGPfact

### Robustness analysis of MGPfact

Before the performance evaluation, we performed a grid search on the number of independent trajectories in 100 training datasets and selected $L = 3$ for downstream testing (***Figure 2—figure supplements 1–2***, Methods).

To further validate the efficacy of MURP downsampling step in MGPfact, we employed an alternative downsampling using randomly selected cells for trajectory inference (Methods). This comparison revealed that the prediction accuracy substantially diminished without MURP, evidenced by a notable reduction in branch assignment ($F1_{branches}$, 20.5%) and cell ordering ($cor_{dist}$, 64.9%) (***Figure 2—figure supplement 3***). In contrast, trajectory predictions utilizing MURP-based downsampling realized an overall score increase of 5.31 to 185%, underscoring the indispensable role of MURP in the trajectory inference capabilities of MGPfact.

Furthermore, we performed a robustness analysis on the topological consistency of the predicted consensus trajectory by comparing prediction results from randomly sampled subsets of the original data. As a result, the consensus trajectory predicted from random subsets by MGPfact retained a high degree of congruence with those from the original datasets (***Figure 2—figure supplement 4***, $HIM_{mean} = 0.686$). This outcome attests to MGPfact's robustness and generalizability under varying data conditions, hence the capability of retrieving conserved bifurcation trajectories in the data.

### MGPfact predicts cellular trajectories

Then, we assessed the performance of MGPfact for prediction of cellular trajectories in 239 test datasets, including 171 synthetic and 68 real datasets, alongside with another seven existing algorithms. For the overall performance score, MGPfact ($Overall_{mean} = 0.534$) ranked second only to TinGa ($Overall_{mean} = 0.563$) and outperformed the rest of six algorithms (***Figure 2a***). Particularly, MGPfact demonstrated the highest accuracy in predicting cell fate in branching trajectory (***Figure 2b***, $F1_{branches_{mean}} = 0.482$). As for other three individual metrics, MGPfact ranked the fourth in $HIM$ ($HIM_{mean} = 0.606$), the sixth in $cor_{dist}$ ($cor_{dist_{mean}} = 0.507$), and the fourth in $wcor_{features}$ ($wcor_{features_{mean}} = 0.712$) (***Figure 2c–e***).

Next, we compared the performance of MGPfact with the other algorithms in nine different trajectory types, respectively, for predicting differentiated cell fate ($F1_{branches}$). As a result, MGPfact significantly outperformed more than half of the algorithms tested in the following trajectory types (T-test p<0.1, ***Table 1***): disconnected graph (N=5), linear (N=5), bifurcation (N=4), multifurcation (N=4), and tree (N=4). As for the other three metrics, MGPfact also showed advantages in $HIM$ in linear (N=5), bifurcation (N=3), convergence (N=3); and in $wcor_{features}$ in multifurcation (N=6), bifurcation (N=5). Nevertheless, MGPfact showed limited predictive performance for $cor_{dist}$ (***Supplementary file 2***).

It is also worth noting that in 68 test datasets of real cell populations, MGPfact ranked the top of all seven algorithms for 'overall score' (***Figure 3a***). As for the individual metric, MGPfact ranked to top for predicting trajectory topology ($HIM_{mean} = 0.721$, ***Figure 3c***); and the second in trajectory branching ($F1_{branches_{mean}} = 0.600$, ***Figure 3b***) with no significant difference with that of the top predictor (scShaper, T-test, p=0.829). These data show that MGPfact can effectively reconstruct the trajectory of cell fate and retrieve relevant biological processes. As for the other metrics in real test datasets, MGPfact was the fifth in the similarity of cell locations ($cor_{dist_{mean}} = 0.46$), and the third in the similarity of gene significance ($wcor_{features_{mean}} = 0.735$). The differences between the metrics of MGPfact and those of the top performers were subtle (***Figure 3d–e***, $\Delta cor_{dist_{mean}} = 0.07$, $\Delta wcor_{features_{mean}} = 0.05$).

We also analyzed three real-world datasets (***Saelens et al., 2019***), each representing a unique topology of trajectory: linear, single bifurcation, and multiple bifurcations. MGPfact excelled in capturing key developmental trajectory with branch points (***Figure 3—figure supplement 1***). In the linear trajectory, MGPfact accurately predicted the absence of bifurcations, aligning well with the ground truth ($overall = 0.871$). For the bifurcation trajectory, MGPfact successfully identified the main bifurcation ($overall = 0.636$). As for the multifurcation trajectory, MGPfact's prediction is also close to the ground truth, as reflected by the overall score ($overall = 0.566$).

In summary, our data suggest that MGPfact is highly efficient in predicting cell fate in branching trajectory ($F1_{branches}$) and topological structure ($HIM$). These capabilities align with the primary objectives of the algorithm, namely, effective identification of the branching events in the development processes of cells. In addition, MGPfact performed better in real datasets, suggesting its robustness

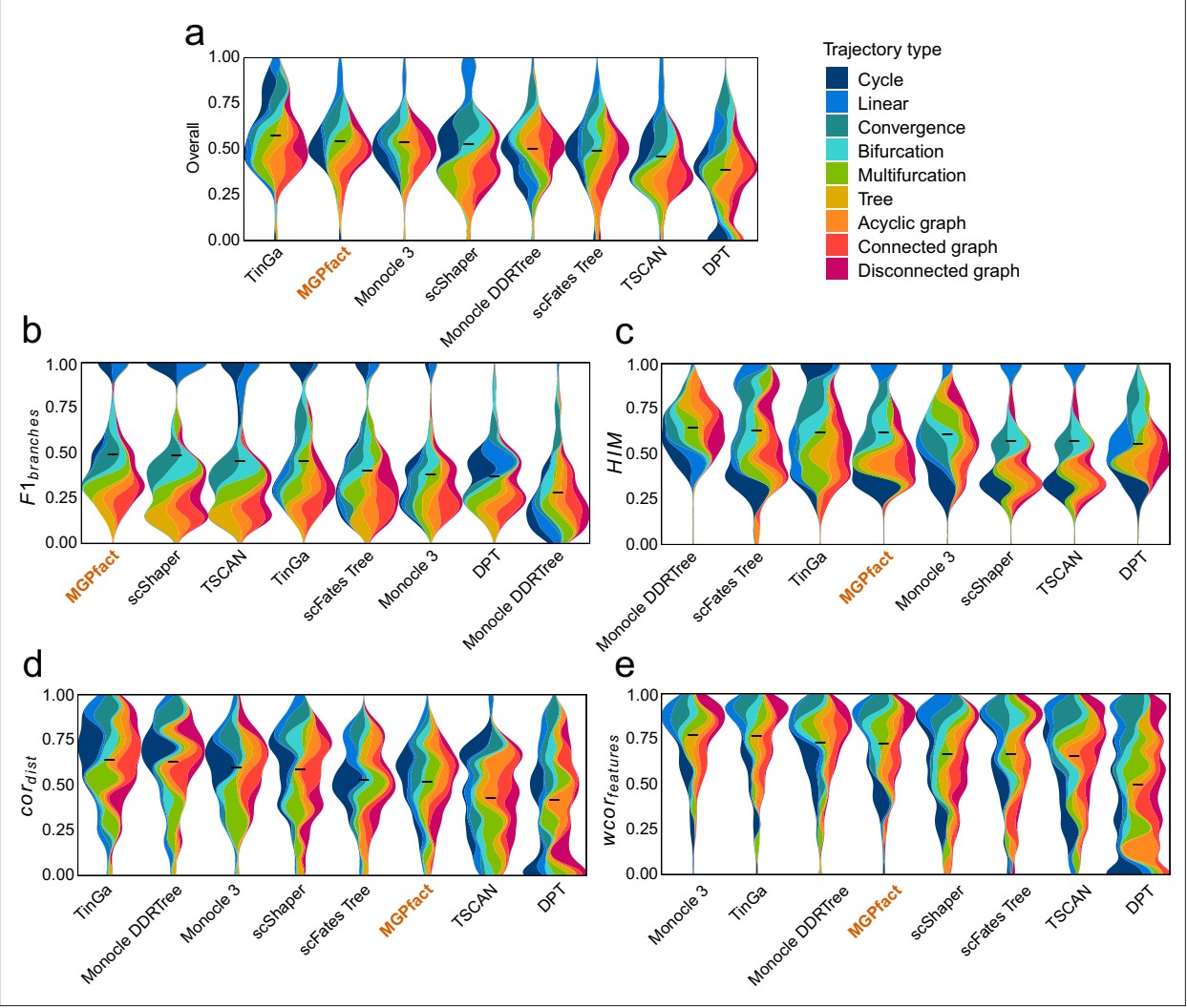

**Figure 2.** Trajectory inference (TI) performance of state-of-the-art methods in 239 test datasets. (**a**) Overall scores; (**b**) $F1_{branches}$; (**c**) $HIM$; (**d**) $cor_{dist}$; (**e**) $wcor_{features}$. All results are color-coded based on the trajectory types, with the black line representing the mean value. The 'Overall' assessment is calculated as the geometric mean of all four metrics.

The online version of this article includes the following figure supplement(s) for figure 2:

**Figure supplement 1.** The distribution of trajectory types among training set and test set.

**Figure supplement 2.** Robustness testing of the number of independent trajectories on 100 training datasets.

**Figure supplement 3.** MURP downsampling enhances the performance of MGPfact.

**Figure supplement 4.** Robustness analysis of consensus trees.

to noise from real experimental conditions. Nevertheless, trajectory inference by MGPfact is based on factorization of the covariance matrix, hence less performance in $wcor_{features}$ and $cor_{dist}$ than methods based on full covariance matrix.

## Comparative of time efficiency and memory consumption

We also compared the runtime and memory usage of different algorithms across 239 test datasets (**Supplementary file 3**). MGPfact's average maximum memory consumption ($memory_{mean}^{(max)} = 0.75 GB$) is comparable to those of the other algorithms ($memory_{mean}^{(max)} \in [0.55, 0.91]\ GB$). As a trade-off for its advantages in feature-selection and factorization, MGPfact requires moderately longer execution time than the other algorithms ($time_{mean} = 3.42 min$).

**Table 1.** MGPfact outperformed state-of-the-art methods in $F1_{branches}$.

*P*-values based one-sided paired t-tests suggest that the $F1_{branches}$ scores of MGPfact were significantly higher than those of the other methods for different trajectory types in the test set.

| Trajectory Type | DPT | Monocle DDRTree | Monocle3 | scFates Tree | scShaper | TinGa | TSCAN |
|---|---|---|---|---|---|---|---|
| Acyclic Graph | 0.169 | 0.735 | 0.341 | 0.167 | **0.006** | 0.485 | **0.002** |
| Bifurcation | 0.559 | **0.001** | **0.015** | 0.298 | **0.010** | 0.772 | **0.002** |
| Convergence | 0.275 | **0.059** | **0.000** | **0.027** | 0.120 | 0.871 | **0.008** |
| Cycle | **0.000** | **0.000** | **0.002** | 0.114 | 0.991 | 0.659 | 0.982 |
| Disconnected Graph | **0.020** | **0.001** | 0.108 | **0.007** | **0.001** | 0.475 | **0.000** |
| Connected Graph | **0.053** | 0.214 | 0.114 | 0.285 | **0.005** | **0.057** | **0.001** |
| Linear | **0.000** | **0.000** | **0.000** | **0.000** | 1.000 | **0.000** | 0.980 |
| Multifurcation | **0.033** | **0.001** | **0.041** | 0.717 | 0.552 | 0.758 | **0.051** |
| Tree | **0.021** | 0.809 | 0.918 | **0.086** | **0.000** | 0.993 | **0.000** |

## MGPfact recovers the trajectory of early postnatal microglia development

The main advantage of MGPfact lies in the capability to factorize a complex cellular trajectory into bifurcation processes of selected co-expressed genes. To illustrate how MGPfact elucidates the biological process underlying cell fate determination, we applied MGPfact to a scRNA-seq data of microglia development and validated the results with experimental evidences (*Li et al., 2019*).

### MGPfact recovers the determinants of the microglia development

Utilizing MGPfact, we reconstructed the developmental trajectories of microglia from immature microglia (IM at pseudotime 0) to homeostatic microglia (HM) and proliferative-region-associated microglia (PAM) (*Figure 4a–c*, left panel). MGPfact identified three bifurcation processes (*Supplementary file 4*), each corresponding to 74–90 highly weighted genes (HWG, absolute gene weight >0.05) (*Figure 4a–c*, right panel).

The first bifurcation determines the differentiated cell fates of PAM and HM, which involves a set of notable marker genes of both cell types, such as Apoe, Selplg (HM), and Gpnmb (PAM). The second bifurcation determines the proliferative status, which is crucial for the development and function of PAM and HM (*Guzmán, 2022*; *Li et al., 2019*). The genes affected by the second bifurcation are associated with cell cycle and proliferation, such as Mki67, Tubb5, Top2a. The third bifurcation influences the development and maturity of microglia, of which the highly weighted genes, such as Tmem119, P2ry12, and Sepp1 are all previously annotated markers for the establishment of the fates of microglia (*Anderson et al., 2022*; *Li et al., 2019*; *Supplementary file 4*).

Moreover, we retrieved highly active regulons from the HWG by MGPfact, of which the significance is quantified by the overall weights of the member genes. These data unveiled highly active transcription regulations in each bifurcation processes, which further traced back to the influential transcription factors as determinants of microglia development, such as Hif1a, E2f5, and Nfkb1 (*Figure 4d–e*, Methods). Specifically, Hif1a is crucial for microglial activation and directly linked to neurodegenerative disease progression (*Wang et al., 2022*). Our data showed an upregulation of Hif1a in the PAM-branch (phase 1) of the first bifurcation, reaffirming the role of Hif1a in PAM differentiation. The other two transcription factors, E2f5 and Nfkb1, were active in phase 2 of the second bifurcation and the third bifurcation, respectively. Both are known for the roles in microglial development (*Dresselhaus and Meffert, 2019*; *Nawal, 2017*).

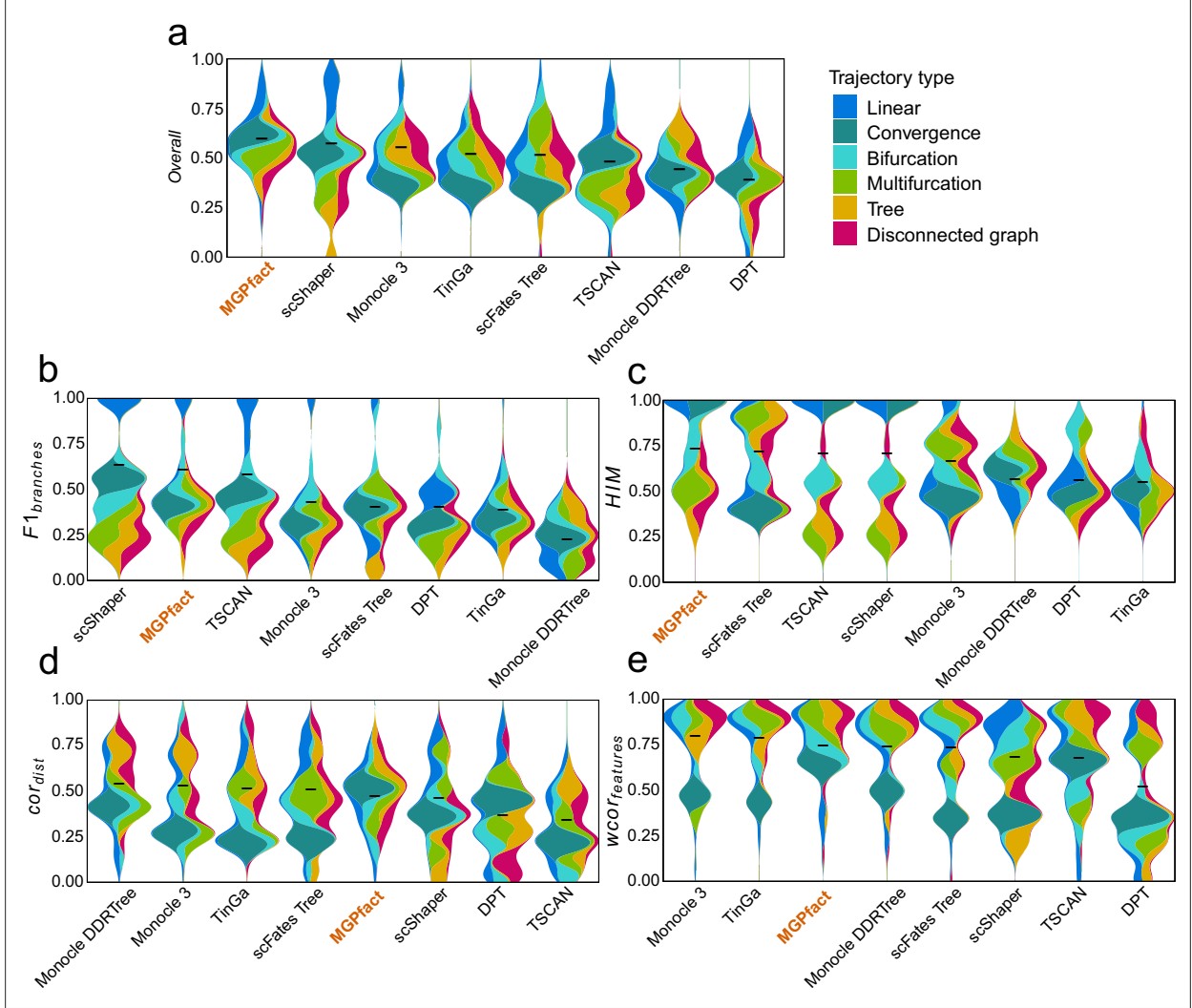

**Figure 3.** Trajectory inference (TI) performance of state-of-the-art methods in 68 test datasets of real cell population. (**a**) Overall scores; (**b**) $F1_{branches}$; (**c**) $HIM$; (**d**) $cor_{dist}$; (**e**) $wcor_{features}$. All results are color-coded based on the trajectory types, with the black line representing the mean value for ranking all methods. The 'Overall' assessment is calculated as the geometric mean of all four metrics.

The online version of this article includes the following figure supplement(s) for figure 3:

**Figure supplement 1.** Trajectories identified by different methods on 3 real-world datasets, with reference structures being linear (**a**) dataset-id=real/silver/germline-human-female_li, bifurcation (**b**) dataset-id=real/silver/fibroblast-reprogramming_treutlein, and multifurcation (**c**) dataset-id=real/silver/oligodendrocyte-differentiation-subclusters_marques.

## Using consensus trajectories to delineate the development of microglial cells

We generated a consensus trajectory of microglial development from three independent bifurcation processes (Methods). Of note, the consensus trajectory revealed two distinct subtypes of proliferative-region-associated microglia (PAM), PAM-T1 (Hif1a+/E2f5-/Nfkb1+), and PAM-T2 (Hif1a+/E2f5-/Nfkb1-) (*Figure 4f*). Particularly, the highly expressed genes in PAM-T2, including Spp1, Gpnmb, Lgals1, and Cd63, are previously identified in disease-associated microglia (DAM) (*Li et al., 2019*). Thus, our finding reaffirmed the connection between the two cell types, DAM and PAM, and suggested Nfkb1 as a potential determinant of differentiation of PAM (*Figure 4g*, *Supplementary file 5*).

In conclusion, MGPfact reconstructed the cellular trajectory of microglial development, identified distinct cell types with marker genes and key regulators which are highly consistent to the experimental evidences (*Dresselhaus and Meffert, 2019*; *Li et al., 2019*; *Nawal, 2017*; *Wang et al., 2022*).

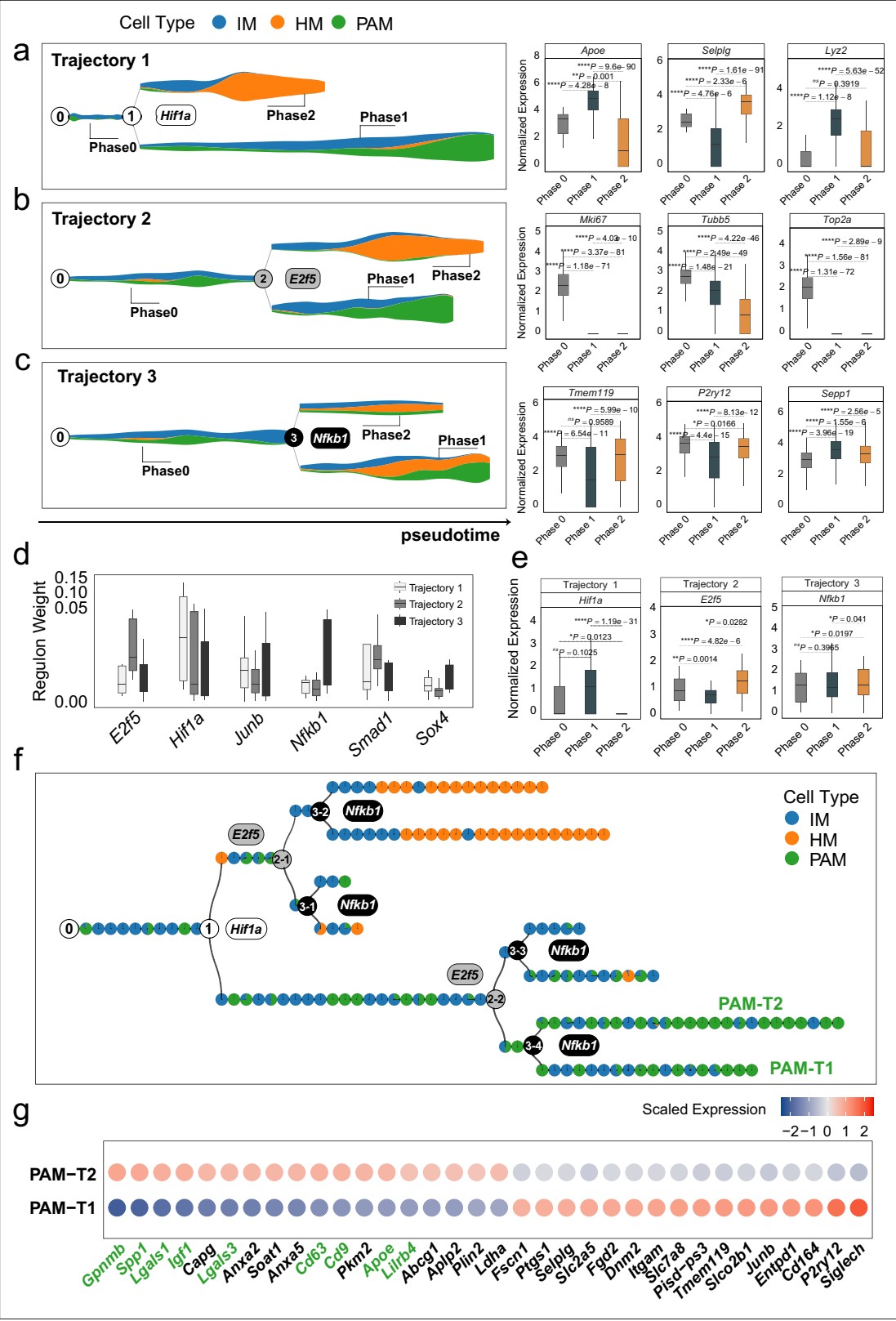

**Figure 4.** MGPfact reconstructed the developmental trajectory of microglia, recovering known determinants of microglia fate. (**a-c**) The inferred independent bifurcation processes with respect to the unique cell types (color-coded) of microglia development, where phase 0 corresponds to the state before bifurcation; and phases 1 and 2 correspond to the states post-bifurcation. Each colored dot represents a metacell of unique cell type defined by MURP. The most highly weighted regulons in each trajectory were labeled by the corresponding transcription factors (left panels). The

*Figure 4 continued on next page*

*Figure 4 continued*

HWG of each bifurcation process include a set of highly weighted genes, of which the expression levels differ significantly among phases 1, 2, and 3 (right panels). (**d**) The most highly weighted regulons influencing the three developmental trajectories of microglia. (**e**) The expression levels of the transcription factors of highly weighted regulons in each trajectory significantly differ among different phases. (**f**) The consensus developmental trajectory by merging the three bifurcation processes. Point 0 denotes the initial of differentiation, whereas the notion of '*n-m*' denotes the *m*-th branch from the branching point *n*. Each colored circle represents a landmark (MURP) of the trajectory, showing the fraction of cell types. The transcription factors of highly weighted regulons in each bifurcation process were used to label each branching point. Particularly, PAM-T1 and PAM-T2 are the two newly defined subtypes of PAM. (**g**) Selected differently expressed genes between PAM-T1 and PAM-T2 (|logfc|>0.25, adjusted *P*-value <0.1) are shown by colored-dots corresponding to the mean expression levels in either cell type. The IDs validated marker genes for PAM are labeled in green. In all box plots, the horizontal line represents the median value, and the whisker extends to the furthest data point within 1.5 times the interquartile range. Significance is denoted as: ns, not significant, * $P<0.05$, ** $P<0.01$, *** $P<0.001$, **** $P<0.0001$, Wilcoxon rank-sum test.

## Using MGPfact to decipher the evolution of tumor-associated CD8[+] T cells

Next, we applied MGPfact to two populations of tumor-associated CD8[+] T cells from non-small cell lung cancer (NSCLC)(*Guo et al., 2018*) and colorectal cancer (CRC) (*Zhang et al., 2018*), respectively. Using the same analytical pipeline as above, we identified a set of CD8[+] T cell gene expression signatures (GES) from MGPfact-inferred trajectories, which are significantly predictive of clinical outcomes and immune treatment responses. Additionally, our data unveiled novel subtypes of tumor-associated CD8[+] T cells with strong clinical implications.

### MGPfact better explains the fate of tumor-associated CD8[+] T cells

We assessed the goodness-of-fit (adjusted R-square) of the consensus trajectory derived by MGPfact and three methods (Monocle 2, Monocle 3, and scFates Tree) for the CD8[+] T cell subtypes described in the original studies (*Guo et al., 2018*; *Zhang et al., 2018*). The data showed that MGPfact significantly improved the explanatory power for most CD8[+] T cell subtypes over Monocle 2, which was used in the original studies (p<0.05, see *Table 2* and *Supplementary file 6*), except for the CD8-GZMK cells in the CRC dataset. Additionally, MGPfact demonstrated better explanatory power in specific cell types when compared to Monocle 3 and scFates Tree. For instance, in the NSCLC dataset,

**Table 2.** Comparison of the explanatory power for CD8[+] T cell fate for MGPfact and three other different methods.

Adjusted R-squared values and P-values based on F-tests demonstrate the relative performance of MGPfact, Monocle 2, Monocle 3, and scFates Tree in fitting the experimentally characterized and annotated CD8[+] T cell subtypes.

| | | MGPfact | | Monocle 2 | | Monocle 3 | | scFates Tree | |
|---|---|---|---|---|---|---|---|---|---|
| | | Adjust R² | p | Adjust R² | p | Adjust R² | p | Adjust R² | p |
| | CD8-LEF1 | **0.935** | 0.000 | 0.176 | 0.000 | 0.089 | 0.08 | 0.902 | 0.000 |
| | CD8-CD28 | **0.195** | 0.002 | 0.170 | 0.000 | 0.108 | 0.06 | 0.006 | 0.145 |
| | CD8-CX3CR1 | 0.634 | 0.000 | 0.259 | 0.000 | 0.629 | 0.000 | **0.882** | 0.000 |
| | CD8-GZMK | 0.259 | 0.000 | 0.189 | 0.000 | **0.855** | 0.000 | 0.547 | 0.000 |
| | CD8-ZNF683 | 0.232 | 0.001 | 0.051 | 0.043 | **0.625** | 0.003 | 0.039 | 0.003 |
| NSCLC (GSE99254) | CD8-LAYN | 0.435 | 0.000 | 0.031 | 0.027 | 0.503 | 0.018 | **0.523** | 0.000 |
| | CD8-LEF1 | 0.311 | 0.000 | 0.027 | 0.036 | 0.461 | 0.007 | **0.99** | 0.000 |
| | CD8-GPR183 | 0.380 | 0.000 | 0.032 | 0.025 | **0.474** | 0.006 | 0.139 | 0.000 |
| | CD8-CX3CR1 | 0.648 | 0.000 | 0.047 | 0.008 | 0.454 | 0.007 | **0.817** | 0.000 |
| | CD8-GZMK | 0.130 | 0.013 | 0.550 | 0.000 | **0.855** | 0.000 | 0.236 | 0.000 |
| | CD8-CD6 | 0.277 | 0.000 | 0.109 | 0.007 | **0.45** | 0.008 | 0.054 | 0.000 |
| | CD8-CD160 | 0.124 | 0.016 | 0.080 | 0.025 | **0.856** | 0.000 | 0.707 | 0.000 |
| CRC (GSE108989) | CD8-LAYN | **0.741** | 0.000 | 0.172 | 0.000 | 0.373 | 0.021 | 0.505 | 0.000 |

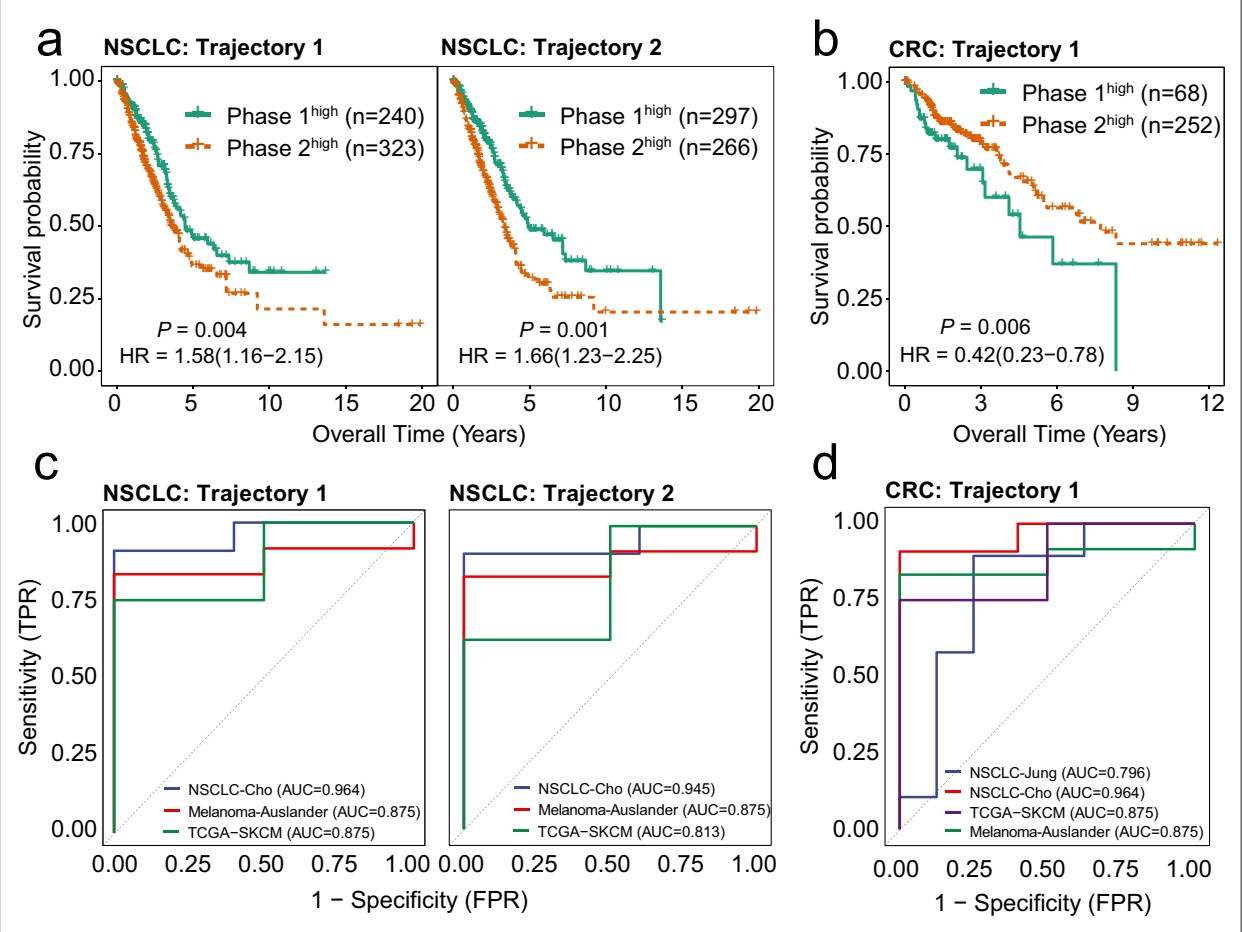

**Figure 5.** Highly weighted genes (HWG) of the bifurcation processes of CD8+ T cells serve as reliable indicators for clinical outcome and immune checkpoint inhibitor (ICI) treatment response. (**a**) Gene expression signatures (GES) corresponding to HWG in CD8+ T cells trajectory 1 and 2 in non-small cell lung cancer (NSCLC) predict overall survival of the TCGA-LUAD cohort. (**b**) GES corresponding to HWG in CD8+ T cells trajectory 1 in colorectal cancer (CRC) predict overall survival of the TCGA-COAD cohort. (**c**) ROC curve showing the weighted mean of HWG in Trajectories 1 and 2 in NSCLC significantly associated with ICI response across three independent studies. (**d**) ROC curve showing the weighted mean of HWG in trajectories 1 and 2 in CRC significantly associated with ICI response across four independent studies.

The online version of this article includes the following figure supplement(s) for figure 5:

**Figure supplement 1.** Differentiation process and determinants of CD8+ T cells from non-small cell lung cancer (NSCLC) environment.

**Figure supplement 2.** Differentiation process and determinants of CD8+ T cells from colorectal cancer (CRC) environment.

**Figure supplement 3.** In non-small cell lung cancer (NSCLC), the weighted mean of highly weighted genes (HWG) from independent differentiation trajectories differs between phases (**a, c**) and immune checkpoint inhibitor (ICI) treatment groups.

**Figure supplement 4.** In colorectal cancer (CRC), the weighted mean of highly weighted genes (HWG) from independent differentiation trajectories differs between phases (**a, c**) and immune checkpoint inhibitor (ICI) treatment groups.

MGPfact exhibited higher explanatory power for CD8-LEF1 cells (*Table 2*, R-squared=0.935), while Monocle 3 and scFates Tree perform better in other cell types.

## MGPfact identifies T-cell gene expression signatures with clinical implications

MGPfact discerned the different cellular fates of tumor-associated CD8+ T cells by distinct bifurcation processes. To reveal the clinical implications of these bifurcation processes, we retrieved the mean expression vectors corresponding to either phase (branch) as GES, and stratified cancer cohorts by quantifying the propensities to certain destiny of CD8+ T cells (Methods). Our data demonstrated pronounced disparities in the clinical outcomes associated with different fates of CD8+ T cells among

patients (*Figure 5a-b*, *Figure 5—figure supplement 2c*). In NSCLC, trajectory 1 is associated with cytotoxic T cells (96%, phase 1) and higher overall survival in TCGA-lung adenocarcinoma (LUAD) patients. Trajectory 2 is associated with exhausted T cells (91%, phase 2), and lower overall survival in the same cohort (*Figure 5a*, *Figure 5—figure supplement 1a-b*). Similarly, in the CRC, trajectory 1 is associated with exhausted T cells (95%, phase 1) and poor overall survival in TCGA-COAD patients (*Figure 5b*, *Figure 5—figure supplement 2a-b*).

In addition, for each trajectory identified in NSCLC and CRC, we selected a set of HWG (absolute gene weight >0.05) to characterize the underlying biological processes and key transcription factors determining the bifurcation (*Supplementary file 7*, *Supplementary file 8*, *Figure 5—figure supplement 1a–b*, *Figure 5—figure supplement 2a–b*). In NSCLC, the HWG of trajectory 1 are primarily implicated in immune responses associated with antigen processing and presentation (*Supplementary file 9*), while trajectory 2-HWG involves processes of immune cell migration (*Supplementary file 9*). For CRC, trajectory 1 is enriched for genes of T cell activation and regulation (*Supplementary file 10*).

Notably, our data showed that the weighted mean expression of the HWG of CD8+ T cell trajectories (Methods) are associated with responses to immune checkpoint inhibitors (ICIs) in multiple independent cohorts (*Figure 5c-d*, *Figure 5—figure supplement 2f*). For instance, the weighted means of HWG of trajectories 1 and 2 in NSCLC which are associated with high activities of cytotoxic T cells, predicted better responses to anti-PD-130 and anti-CTLA-4 (*Cho et al., 2020*; *Liu et al., 2018*), as well as their combination therapies (*Auslander et al., 2018*) ($AUC \in \{0.813, 0.964\}$, $P<0.1$, *Figure 5—figure supplement 3*). Similarly, HWG pertaining to trajectory 1 in CRC is associated with high proportion of *EMRA*(87%, phase 1), hence better responses to immunotherapies in 4 cohorts (*Auslander et al., 2018*; *Cho et al., 2020*; *Jung et al., 2019*; *Liu et al., 2018*) ($AUC \in \{0.796, 0.964\}$, p<0.01, *Figure 5—figure supplement 4*).

Taken together, the factorization of scRNA-seq data by MGPfact provides highly relevant gene expression signatures of the fate of tumor-associated CD8+ T cells, which advances the understanding of the evolution of tumor immune microenvironment (TIME) and predicts clinical outcomes.

## MGPfact identified new subtypes of CD8+ T cells with clinical implications

Furthermore, the consensus trajectories of tumor-associated CD8+ T cells inferred by MGPfact from NSCLC and CRC revealed new subtypes of lymphocytes. In NSCLC, we characterized CD8-ZNF683-T1 (LEF+/TBX21-) and CD8-ZNF683-T2 (LEF+/TBX21+) from CD8-ZNF683 (*Figure 6a*). The CD8-ZNF683-T2 cells highly expressed genes associated with 'pre-exhausted' state, such as ITGAL, SAMD3, and SLAMF7 (*Pritchard et al., 2023*), many of which are target genes of TBX21. In contrast, CD8-ZNF683-T1 showed lower expression of these genes, hence repellency to the 'pre-exhausted' state (*Figure 6b*, *Supplementary file 11*). In CRC, we identified two subtypes of effector memory T cells (CD8-GZMK), CD8-GZMK-T1 (EOMES-/BHLHE40+), and CD8-GZMK-T2 (EOMES-/BHLHE40-) (*Figure 6d*). CD8-GZMK-T2 strongly resembles CD8-GZMK, and potentially differentiating into CD8-CD6 (resident memory T cells, $T_{RM}$) and CD8-CD160 (intraepithelial lymphocytes, *IEL*); whereas CD8-GZMK-T1 cells demonstrated higher expression of GZMB, indicating an active cytotoxic cell-killing capability (*Trapani, 2001*). Simultaneously, these cells also marked by high expression levels of immune related genes, including OASL, RBPJ, and CTLA4, which are known targets of BHLHE40 (*Lutter et al., 2022*; *Salmon et al., 2022*), implying that BHLHE40 is a modulator of the higher effector activity in CD8-GZMK-T1 (*Figure 6e*, *Supplementary file 12*).

We further derived scores based on the differentially expressed genes of CD8-ZNF683-T1 and CD8-GZMK-T1 (Methods), as measures of the fraction of each subtype in cancer cohorts. In the LUAD cohort of TCGA, increased fraction of CD8-ZNF683-T1 in TIME was associated with favorable outcomes (*Figure 6c*). And increased fractions of CD8-GZMK-T1 in TIME were associated with better responses to ICI therapy across three independent cohorts[31–33] (*Figure 6f*, *Figure 6—figure supplement 1*, $AUC \in \{0.782, 0.917\}$, p<0.1), which were treated with anti-PD-1, anti-CTLA-4 and their combination therapies.

In conclusion, our data showed the cellular trajectory inferred by MGPfact can be used to elucidate the complex evolutionary processes of tumor-associated CD8+ T cells, and further inform the characterization of new subtypes of T cells with significant clinical implications.

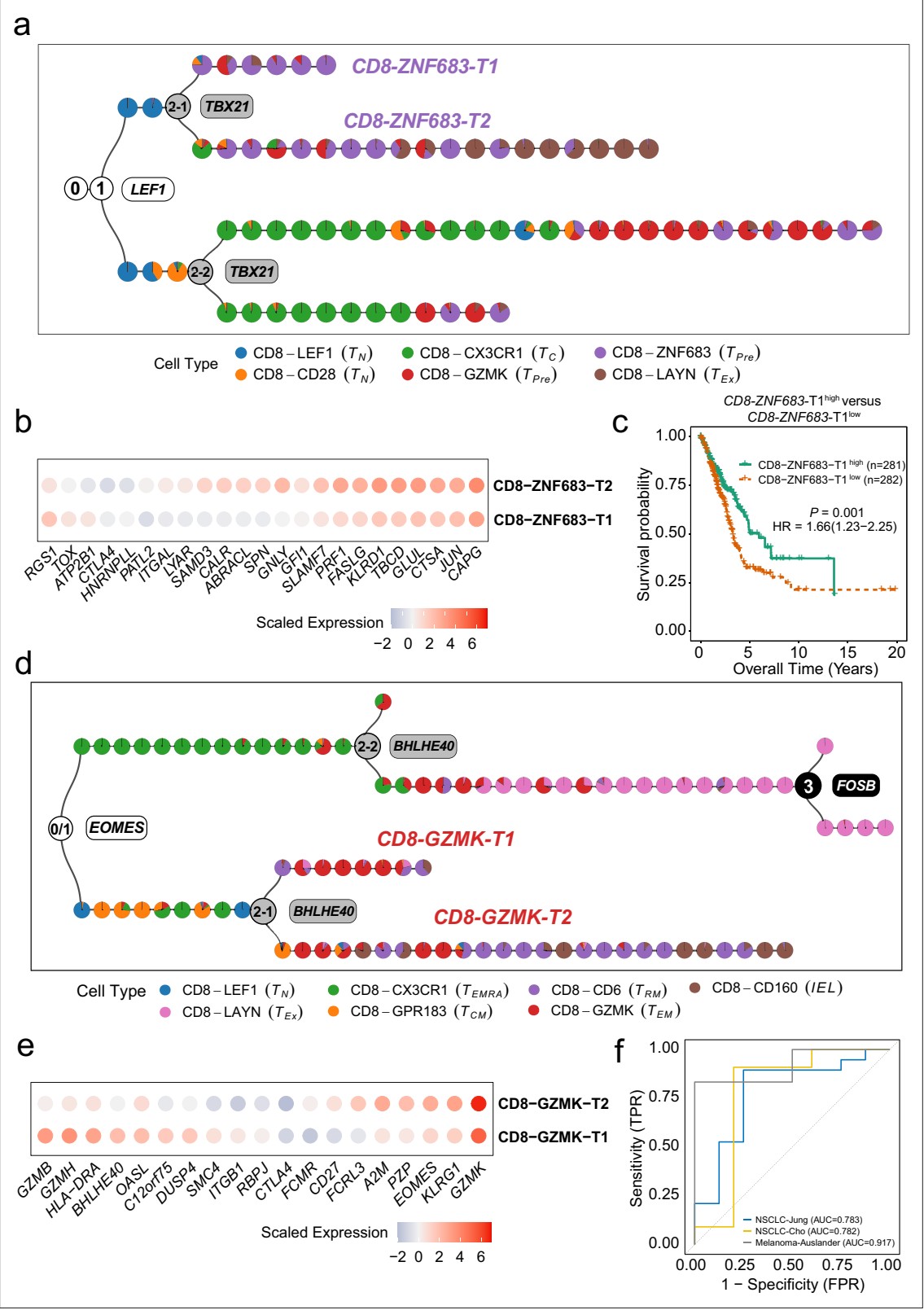

**Figure 6.** MGPfact serves as an effective approach for characterization of new cellular subtypes. (**a**) The consensus trajectory of tumor-associated CD8⁺ T cells in non-small cell lung cancer (NSCLC) identified CD8-ZNF683-T1 and CD8-ZNF683-T2 as two subtypes of CD8-ZNF683, which are influenced by TBX21. (**b**) Selected differently expressed genes between CD8-ZNF683-T1 and CD8-ZNF683-T2 (|logfc|>0.25, adjusted p-value <0.1). (**c**) High expression of CD8-ZNF683-T1 signatures predicts good overall survival in the TCGA LUAD cohort (Methods). p-values were calculated through multivariate Cox

*Figure 6 continued on next page*

*Figure 6 continued*

regression analysis, and HR represents hazard ratio. (**d**) The consensus trajectory of tumor-associated CD8$^+$ T cells in colorectal cancer (CRC) identified CD8-GZMK-T1 and CD8-GZMK-T2 as two subtypes of CD8-GZMK. (**e**) Selected differently expressed genes between CD8-GZMK-T1 and CD8-GZMK-T2 (|logfc|>0.25, adjusted p-value <0.1). (**f**) ROC curve showing high expression of CD8-GZMK-T1 signature associated with ICI treatment response in three independent studies. The consensus trajectory is formed by merging three bifurcation processes. Each colored circle represents a landmark (minimum unbiased representative points, MURP), indicating the cell type.

The online version of this article includes the following figure supplement(s) for figure 6:

**Figure supplement 1.** The CD8-GZMK-T1 scores between immune checkpoint inhibitor (ICI) treatment response groups (Methods).

## Discussion

scRNA-seq provides a direct, quantitative snapshot of a population of cells in certain biological conditions, thereby revealing the actual cell states and functions. Although existing clustering and embedding algorithms can effectively reveal discrete biological states of cells, these methods become less efficient when depicting continuous evolving of cells over the temporal domain. The introduction of manifold learning offers a new dimension for discovery of relevant biological knowledge in cell fate determination, allowing for a better representation of continuous changes in cells, especially in time-dependent processes such as development, differentiation, and clonal evolution. However, current manifold learning methods face major limitations, such as the need for prior information on pseudo-time and cell clustering, and lack of explainability, which restricts their applicability. Additionally, many existing trajectory inference methods do not support gene selection, making it difficult to annotate the results to known biological entities, thereby hindering the interpretation of results and subsequent functional studies.

We developed MGPfact to overcome the limitations of the existing methods. Inspired by recent studies, MGPfact model the cell fate as mixture of Gaussian processes, which accommodate both continuous evolution pathway and biphasic destiny of cell fate. Thus, MGPfact is capable to distinguish discrete and continuous events in the same trajectory. In addition, by factorizing the mixture of Gaussian processes, MGPfact offers the advantage to select genes corresponding to each bifurcation process and thereby enables full biological annotation and interpretation of the trajectory. As a validation, we showed that gene-selection by MGPfact consistently recapitulated the development of microglia and tumor-associated CD8$^+$ T cells; and recovered key regulators of distinct cell fate. So far, MGPfact is the only model-based manifold-learning framework which factorizes complex development trajectories into independent bifurcation processes of gene sets.

We conducted a comprehensive comparison of MGPfact with existing TI methods from various perspectives. This comparison included the correlation of cell sorting, accuracy of branch allocation, similarity of topological structures, and differentially expressed features. It ought to be mentioned that number of principal components used should be determined by the intrinsic biological characteristics of the cell fate-determination. Our experiment based on a limited number of datasets may not represent more complex scenarios in other cell types. For the overall TI-performance, MGPfact demonstrated leading performance across 239 datasets, second only to TinGa. For the performance in branch allocation, which directly reflects the fitness to the outcomes of cell-fate, MGPfact outperformed its counterparts, especially in the topology groups of linear and bifurcation. As for $wcor_{features}$ and $cor_{dist}$, MGPfact performed less well mainly for two reasons. First, MGPfact is designed for bifurcation topology in the cellular trajectory, hence less efficient in inferring complex topologies. Then, MGPfact inference is based on selected gene sets instead of the whole transcriptome, the resulted trajectory correspond only to the bifurcation processes of interest and hence does not necessarily reflect the whole topology of cellular trajectory. Furthermore, MGPfact performed significantly better in trajectory prediction in real cell population compared to the synthetic ones, suggesting the algorithm fit better to the true biological variation and noises.

To reconstruct the trajectory of cell fate, we merged all the bifurcation processes into a consensus trajectory. In the validation by microglia and tumor-associated CD8$^+$ T cells, the consensus trajectory revealed highly consistent findings recovering known biology and the marker genes of specific cell states which further inform the transcription factor (TF) determining the fate of cell. In addition, the consensus trajectory revealed new subtypes of cells demonstrating highly relevant transcriptional characteristics. Particularly, we reported new subtypes of tumor-associated CD8$^+$ T cells characterized

by different TBX21 and BHLHE40 activity, both of which are known regulators of CD8$^+$ T cell functionality (*Lutter et al., 2022*; *Pritchard et al., 2023*; *Salmon et al., 2022*; *Trapani, 2001*). These data suggest that MGPfact is capable to discover gene modules with strong and consistent transcriptional background hence better interpretability. Moreover, the results of MGPfact demonstrated strong clinical relevance. Using MGPfact we retrieved GES which quantitatively measure the propensity and fraction of different fates of CD8$^+$ T cell. These signatures correspond to important biological processes of T cell activity; and predict clinical outcome and ICI treatment responses from transcriptome data of bulk tumor biopsies, independent of any endogenous feature of the tumor cells.

Nevertheless, MGPfact also has some limitations, which shall be addressed in future study. Firstly, the complex definition of the bifurcation kernel introduces unfavorable singularity to the Gaussian kernel when considering highly complex trajectories. Additionally, the current trajectory inference by MGPfact is solely based on the temporal domain, neglecting bifurcation processes occurring in space. To overcome these limitations, future models should incorporate spatial dynamics of transcription and RNA velocity data to provide more comprehensive insights of cell fate. Moreover, the reconstruction of cellular trajectories by MGPfact implies independence of each bifurcation processes, which may not reflect real cellular behavior. Therefore, predictions from MGPfact should be interpreted with caution and validated experimentally.

## Methods
### Benchmarking MGPfact to state-of-the-art methods

We adopted a comprehensive evaluation framework from previous scRNA-seq study to assess the TI performance of MGPfact (*Saelens et al., 2019*; *Smolander et al., 2022*; *Todorov et al., 2020*). The validation dataset comprises 110 real data and 229 synthetic data, encompassing nine different cellular trajectory topologies. The ground truth of cellular trajectories of each dataset were inferred and validated by the original study (*Saelens et al., 2019*). The synthetic datasets were generated using four simulators: dyngen (*Saelens et al., 2019*), dyntoy (*Saelens et al., 2019*), PROSSTT (*Papadopoulos et al., 2019*), and Splatter (*Zappia et al., 2017*), each modeling different trajectory topologies such as linear, branching, and cyclic. Splatter simulates branching events by setting expression states and transition probabilities, dyntoy generates random expression gradients to reflect dynamic changes, and dyngen focuses on complex branching structures within gene regulatory networks.

The evaluation of TI performance was based on five metrics.

1. The Hamming-Ipsen-Mikhailov (HIM) distance is a metric for assessing similarity between two topological structures. It integrates the normalized Hamming distance, which highlights differences in edge lengths, with the Ipsen-Mikhailov distance, which focuses on similarities in degree distributions. By linearly combining these two measures, the HIM distance offers a comprehensive evaluation of both local and global structural differences.
2. The $F1_{branches}$ is a metric used to evaluate a model's accuracy in branch allocation. It represents the harmonic mean of precision and recall, effectively capturing the performance of branch identification. In trajectory inference, $F1_{branches}$ are calculated by assessing the similarity between predicted and actual trajectory branches, emphasizing the Jaccard similarity of branch pairs.
3. This $cor_{dist}$ metric measures similarity in intercellular distances between predicted and actual trajectories. It evaluates model accuracy in cell ordering by comparing relative positions of paired cells, highlighting changes in cell positions across states, and reflecting cell differentiation dynamics.
4. The $wcor_{features}$ metric evaluates the similarity of key features, such as differentially expressed genes, between predicted and actual trajectories. Using weighted Pearson correlation, it highlights consistent features, reflecting the model's ability to capture biological variation. This helps identify crucial genes in trajectories and understand the molecular mechanisms of cell state transitions.
5. The *overall* score is calculated by taking the geometric mean of the four aforementioned metrics, providing an assessment of overall performance.

The dataset is divided into two groups: a training set and a testing set (*Figure 2—figure supplement 1*). We use 100 training datasets to perform the following tasks:

1. Determine the optimal number of trajectories: With three set as the default for the number of factorized trajectories, we tested other values (1, 2, 4, and 5) and used paired T-tests to assess

whether there are significant changes in MGPfact's prediction results under different parameter settings.

2. Verify the critical role of MURP: Randomly select 20, 40, 60, 80, and 100 cells for trajectory inference, map the inference results back to the original data using the KNN graph structure, and compare the prediction results with those obtained through MURP downsampling.

3. Robustness analysis of the consensus trajectory topology: Perform 60%, 70%, 80%, and 90% sampling on the original data, and then calculate the similarity between the consensus trajectory predictions of MGPfact with and without sampling. A higher score indicates better robustness of the method.

Subsequently, in 239 test datasets, we compare MGPfact with seven state-of-the-art TI methods using the aforementioned metrics, including Monocle DDRTree (*Qiu et al., 2017b*, *Qiu et al., 2017a*), TSCAN (*Ji and Ji, 2016*), and DPT (*Haghverdi et al., 2016*), as well as four new methods from recent studies: Monocle 3 (*Cao et al., 2019*), scShaper (*Smolander et al., 2022*), scFates Tree (*Faure et al., 2023*), and TinGa (*Todorov et al., 2020*).

The experimental comparisons were conducted on a CentOS system equipped with 48 CPU cores running at 2.2 GHz and 250 GB of memory. To ensure a uniform comparison, all experiments were performed using a single CPU core. For MGPfact, we tested each resulted trajectory and selected the one with the best 'overall' score for comparison. For the other seven methods, default settings are used unless otherwise specified.

## Application of MGPfact in a microglia single-cell RNA-seq dataset

We utilized the MGPfact reconstructed the developmental trajectory of microglia from a scRNA-seq dataset, including IM, PAM, and HM (*Li et al., 2019*). In this analysis, we provided a detailed explanation of the analytical steps of MGPfact and the key results. First, we identified three independent developmental pathways and pinpointed HWG associated with each bifurcation process. Then, we retrieved highly active regulons within each bifurcation process, tracing back to the potential influential determinants (transcription factors) in the development of microglia. Finally, we combined all the bifurcation processes into a consensus trajectory, which recovered the known biology of disease-related microglia (PAM), as represented by distinct cellular states and marker genes.

## Predicting the evolutionary trajectory of tumor-associated CD8$^+$ T cells

We utilized MGPfact to conduct an exploratory analysis of the evolution of tumor-associated CD8$^+$ T cells of non-small cell lung cancer (NSCLC) and colorectal cancer (CRC). We evaluated the goodness-of-fitness of the consensus trajectories from MGPfact to the CD8$^+$ T cell subtypes identified in the original studies. For comparison, we used Monocle 2 (*Qiu et al., 2017b*, *Qiu et al., 2017a*) as a baseline model.

For the survival analysis, we extracted GES from each independent bifurcation process to develop classifiers for evolutionary propensity of CD8$^+$ T cells towards specific fates, based on which we stratified TCGA cancer cohorts and verified their association with clinical outcome.

To evaluat the association to ICI responses, we used HWG to retrieve key transcription factors related to each bifurcation process and characterized the underlying biological processes. We then assessed their connection to immunotherapy response using weighted means of the HWG.

Finally, we identified new subtypes based on distinct endpoints of the consensus trajectory and validated their association with clinical outcome and immunotherapy response using the mean of the differently expressed gene (DEG).

## Single-cell sequencing data processing

We obtained the original mouse developmental microglia single-cell sequencing data from the GEO accession number GSE123025 (*Li et al., 2019*). Using Seurat (*Butler et al., 2018*; *Stuart et al., 2019*), we replicated the processing steps described in the original study: (1) Normalization by dividing gene expression values by total RNA count, followed by log2 transformation; (2) Selection of highly variable genes (HVGs) using Seurat's mean.var.plot function, with controlled average expression $[0.0125, 3]$ and variance $[0.5, \infty]$; (3) Scaling and centering of the normalized matrix for HVGs, with regression of cell cycle effects. After preprocessing, we grouped cells into IM (P7-C0), PAM (P7-C1 and P7-GPNMB$^+$-CLEC7A$^+$), and HM (P60), resulting in a 4889-gene expression matrix across 1009 cells.

For analyzing tumor-associated CD8[+] T cells, we utilized scRNA-seq data from lung cancer (GSE99254) (*Guo et al., 2018*) and colorectal cancer (GSE108989) (*Zhang et al., 2018*) in the GEO database. We extracted preprocessed and centralized gene expression matrices of CD8[+] T cells and analyzed them using the same genes and the same method (Monocle 2) as in the original papers or MGPfact for trajectory construction for a direct comparison. The NSCLC data yielded an 888-gene expression matrix across 3700 cells, while the CRC data resulted in a 700-gene expression matrix across 3177 cells.

## Functional enrichment of highly weighted genes

For the highly weighted genes (HWG, absolute gene weight >0.05) obtained from independent bifurcation processes, we utilized the R package clusterprofiler (*Yu et al., 2012*) to perform functional annotation using GO terms (*Harris et al., 2004*), including biological process (BP), cellular component (CC), and molecular function (MF). The results with a Benjamini–Hochberg-adjusted p-value less than 0.05 were retained.

## Transcription factor program analysis

To comprehensively assess key regulatory factors within each independent trajectory, we performed SCENIC (*Aibar et al., 2017*) transcription factor regulatory program estimation for each analysis case. GENIE3 (*Huynh-Thu et al., 2010*) was used to identify co-expressed modules from the results of MURP (*Ren et al., 2022*) downsampling. RCisTarget (*Aerts et al., 2010*; *Aibar et al., 2017*) was then used to identify regulons before AUCell (*Aibar et al., 2017*) was used to estimate the activity of each regulon. Each regulon comprises a specific transcription factor and its target genes. Finally, we utilize gene weights obtained from MGPfact analysis to evaluate the distinct impact of top regulons on each trajectory.

## Generating the consensus trajectory

Following MGPfact decomposition, we obtained multiple independent bifurcation trajectories, each corresponds to a binary tree within the temporal domain. These trajectories were then merged to construct a coherent diffusion tree, representing the consensus trajectory of cells' fate. The merging process involves initially sorting all trajectories by their bifurcation time. The first (earliest) bifurcation trajectory is chosen as the initial framework, and subsequent trajectories are integrated to the initial framework iteratively by adding the corresponding branches at the bifurcation timepoints. As a result, the trajectories are ultimately merged into a comprehensive binary tree, serving as the consensus trajectory.

## Assessing consistency of MGPfact-derived CD8[+] T cell subtype trajectories

In the case study of CD8[+] T cells, by combining independent trajectories, we derive a consensus trajectory representing the complex developmental pathway. To assess the goodness-of-fitness to the CD8[+] T cell subtypes from the original study, we classified the trajectories into several states based on bifurcation points, each corresponding to a distinct stage of the evolutionary process. Then, we evaluated the interactive effects between the states and pseudotime on the fraction of the cell types using F-test (ANOVA). The resulted R-squared (R2), p-values, and F-statistics were used to evaluate the goodness-of-fitness of the models tested hence the explanatory power. For comparison, we used the Monocle 2 as the baseline model for trajectory inference. The differentiation trajectories of Monocle 2 were replicated following the workflow in the original study (*Guo et al., 2018*; *Zhang et al., 2018*).

## Survival analysis

We assessed the association of bifurcation processes and specific cell types with the clinical outcomes two cohorts of lung adenocarcinoma (LUAD, N=563) and colon adenocarcinoma (COAD, N=320) data from The Cancer Genome Atlas (TCGA). The gene expression and clinical data were downloaded from UCSC Xena platform (http://xena.ucsc.edu/).

We assessed the survival impacts of NSCLC and CRC bifurcation processes in the TCGA LUAD and COAD cohorts, respectively. For each independent bifurcation process, we defined GES by the mean expression vectors of all trajectories in phase 1 and 2, respectively. Subsequently, we calculated

the Pearson's correlation coefficients for each individual expression profile in TCGA LUAD or COAD cohorts to the phase 1 and 2 GES, where higher correlation corresponds to a stronger propensity to specific cell fate. This allowed us to classify patients into two groups: those exhibiting propensity to phase 1 and those exhibiting propensity to phase 2.

To assess the survival impacts of specific cell states defined by the consensus trajectory, we developed a CD8-ZNF683-T1 score based on the signed average expression level of DEG associated with CD8-ZNF683-T1. Subsequently, we classified the TCGA LUAD cohorts into two groups using the median of CD8-ZNF683-T1 scores, identifying those demonstrating a propensity towards CD8-ZNF683-T1 and those demonstrating a propensity towards CD8-ZNF683-T2.

To adjust for possible confounding effects, the relevant clinical features including age, sex, and tumor stage were used as covariates. The Cox regression model was implemented using R-4.2 package 'survival.' And, we generated Kaplan-Meier survival curves based on different classifiers to illustrate differences in survival time and report the statistical significance based on Log-rank test.

## Immune-checkpoint inhibitor treatment response analysis

For the prediction of Immune-checkpoint inhibitor treatment response, we collected four datasets containing ICI treatment responses. These datasets consist of two non-small cell lung cancer-related datasets (GSE135222 [*Jung et al., 2019*], n=27; GSE126044 [*Cho et al., 2020*], n=16) and two melanoma-related datasets (GSE115821 [*Auslander et al., 2018*], n=14; TCGA-MENO [*Liu et al., 2018*], n=10). All data were processed with DESeq2 (*Love et al., 2014*) to fit gene dispersion to a negative binomial distribution, normalize raw counts, and stabilize variance, achieving standardization.

To validate the bifurcation processes identified by MGPfact predicting patients' response to immune-checkpoint inhibitor (ICI) treatments, we selected highly weighted genes (HWG) with absolute weights greater than 0.05 from each independent bifurcation process. Then, we calculated the weighted mean expression of HWG in each ICI dataset to generate ROC curves for patient drug response.

To validate the specific cell states defined by the consensus trajectory predicting patients' response to ICI treatments, we used CD8-GZMK-T1 score based on the average expression level differences of upregulated and downregulated genes associated with CD8- GZMK-T1. Then, we calculated the CD8-GZMK-T1 score in each ICI dataset to generate ROC curves for patient drug response.

## Acknowledgements

We would like to thank Qingyun Li for providing the cell labels for the microglial dataset (*Li et al., 2019*). We would also like to express our gratitude to Nengming Xiao, and Guo Fu for their valuable input and constructive suggestions during the preparation of the manuscript. This work was supported by the Fundamental Research Funds for the National Natural Science Foundation of China [82272944 to QL]; the National Natural Science Foundation of China [82203420 to JG]; the National Key Research and Development Program of China [2022YFC2704801 to QL].

## Additional information

### Funding

| Funder | Grant reference number | Author |
| --- | --- | --- |
| National Natural Science Foundation of China | 82272944 | Qiyuan Li |
| National Natural Science Foundation of China | 82203420 | Jintao Guo |
| National Key Research and Development Program of China | 2022YFC2704801 | Qiyuan Li |

The funders had no role in study design, data collection and interpretation, or the decision to submit the work for publication.

## Author contributions
Jun Ren, Data curation, Software, Formal analysis, Validation, Investigation, Visualization, Methodology, Writing – original draft, Writing – review and editing; Ying Zhou, Jing Yang, Hongkun Fang, Xuejing Lyu, Jintao Guo, Xiaodong Shi, Writing – review and editing; Yudi Hu, Methodology; Qiyuan Li, Supervision, Methodology, Project administration, Writing – review and editing

## Author ORCIDs
Jun Ren https://orcid.org/0000-0002-9027-2303
Qiyuan Li https://orcid.org/0000-0002-8934-8948

Reviewer #1 (Public review): https://doi.org/10.7554/eLife.97424.3.sa1
Reviewer #2 (Public review): https://doi.org/10.7554/eLife.97424.3.sa2
Author response https://doi.org/10.7554/eLife.97424.3.sa3

## Additional files

### Supplementary files
Supplementary file 1. The specific configuration of parameters contained in the model.

Supplementary file 2. p-values associated with one-sided paired t-tests assess whether the performance score (across five metrics) of MGPfact is significantly higher than the other methods on the different trajectory types in test set.

Supplementary file 3. Comparison of memory utilization (GB) and temporal efficiency (min).

Supplementary file 4. The highly weighted genes associated with independent bifurcating processes in microglia development (absolute gene weight >0.05).

Supplementary file 5. List of genes specifically expressed in homeostatic microglia (HM). PAM-T1 cells (n=113) vs. PAM-T2 cells (n=298), two-sided moderated t-test with limma.

Supplementary file 6. Comparison of the explanatory power for CD8+ T cell fate for MGPfact and three other different methods. Adjusted R-squared values and p-values based on F-tests demonstrate the relative performance of MGPfact, Monocle 2, Monocle 3, and scFates Tree in fitting the experimentally characterized and annotated CD8+ T cell subtypes.

Supplementary file 7. The highly weighted genes associated with the CD8+ T cell-independent bifurcating processes in non-small cell lung cancer (NSCLC) (absolute gene weight >0.05).

Supplementary file 8. The highly weighted genes associated with the CD8+ T cell-independent bifurcating processes in colorectal cancer (CRC) (absolute gene weight >0.05).

Supplementary file 9. GO Enrichment Analysis Results for Highly Weighted Genes Associated with Trajectory 1 of non-small cell lung cancer (NSCLC) (Benjamini–Hochberg-adjusted p-value <0.05).

Supplementary file 10. GO Enrichment Analysis Results for Highly Weighted Genes Associated with Trajectory 1 of colorectal cancer (CRC) (Benjamini–Hochberg-adjusted p-value <0.05).

Supplementary file 11. List of genes specifically expressed in CD8-ZNF683. CD8-ZNF683-T1 cells (n=204) vs. CD8-ZNF683-T2 cells (n=395), two-sided moderated t-test with limma.

Supplementary file 12. List of genes specifically expressed in CD8-GZMK. CD8-GZMK-T1 cells (n=155) vs. CD8-GZMK-T2 cells (n=221), two-sided moderated t-test with limma.

MDAR checklist

### Data availability
The datasets used for method performance comparison are archived on Zenodo platform, with processed real and synthetic datasets available at https://doi.org/10.5281/zenodo.1443566. All instance data used in this article can be downloaded from the GEO database with accession numbers GSE123025, GSE99254, and GSE108989.The expression matrices related to immune- checkpoint inhibitors (ICI) and clinical response information used in this article were downloaded from the GEO database (GSE135222; GSE126044, GSE115821) and TCGA (TCGA-SKCM). This is a computational study, in which no new experimental data were generated. We have developed a comprehensive workflow for MGPfact. Firstly, a Docker container enables one-click program execution (details at: https://github.com/renjun0324/ti_mgpfact, copy archived at *Ren, 2025a*). Additionally, to fully

harness MGPfact's capabilities, we have created the R package MGPfactR, accessible at: https://github.com/renjun0324/MGPfactR (copy archived at *Ren, 2025b*). Within this workflow, MCMC sampling for model parameter estimation is carried out using the Mamba library in the Julia environment. The related Julia package can be found here: https://github.com/renjun0324/MGPfact.jl (copy archived at *Ren, 2025c*). Other analysis scripts can be found on GitHub at https://github.com/renjun0324/mgpfact_paper (copy archived at *Ren, 2025d*). And the scFates Tree used in this paper is available as a performance comparison Docker container, constructed using the dendritic trajectory process in scFates, accessible at: https://github.com/renjun0324/ti_scfates_tree (copy archived at *Renjun0324, 2025*).

The following previously published datasets were used:

| Author(s) | Year | Dataset title | Dataset URL | Database and Identifier |
|---|---|---|---|---|
| Guo X, Zhang Y, Zheng L, Zheng C, Song J, Zhang Q, Kang B, Liu Z, Jin L, Xing R, Gao R, Zhang L, Dong M, Hu X, Ren X, Kirchhoff D, Roider HG, Yan T, Zhang Z | 2018 | T cell landscape of non-small cell lung cancer revealed by deep single-cell RNA sequencing | https://www.ncbi.nlm.nih.gov/geo/query/acc.cgi?acc=GSE99254 | NCBI Gene Expression Omnibus, GSE99254 |
| Li Q, Cheng Z, Zhou L, Darmanis S, Neff NF, Okamoto J, Gulati G, Bennett ML, Sun LO, Clarke LE, Marschallinger J, Yu G, Quake SR, Wyss-Coray T, Barres BA | 2018 | Deep single-cell RNAseq of microglia and brain myeloid cells from various brain regions and developmental stages | https://www.ncbi.nlm.nih.gov/geo/query/acc.cgi?acc=GSE123025 | NCBI Gene Expression Omnibus, GSE123025 |
| Saelens W, Cannoodt R, Todorov H, Saeys Y | 2019 | Single-cell -omics datasets containing a trajectory | https://doi.org/10.5281/zenodo.1443566 | Zenodo, 10.5281/zenodo.1443566 |
| Jung H, Kim HS, Kim JY, Sun JM, Ahn JS, Ahn Mj, Park K, Esteller M, Lee SHJ, Choi JK | 2019 | DNA methylation loss coupled with mitotic cell division promotes immune evasion of tumours with high mutation load [RNA-seq] | https://www.ncbi.nlm.nih.gov/geo/query/acc.cgi?acc=GSE135222 | NCBI Gene Expression Omnibus, GSE135222 |
| Auslander N, Zhang G, Lee JS, Frederick DT, Miao B, Moll T, Tian T, Wei Z, Madan S, Sullivan RJ, Boland G, Flaherty K, Herlyn M, Ruppin E | 2018 | Robust prediction of response to immune checkpoint blockade therapy in metastatic melanoma | https://www.ncbi.nlm.nih.gov/geo/query/acc.cgi?acc=GSE115821 | NCBI Gene Expression Omnibus, GSE115821 |
| Zhang L, Yu X, Zheng L, Zhang Y, Li Y, Fang Q, Gao R, Kang B, Zhang Q, Huang JY, Konno H, Guo X, Ye Y, Gao S, Wang S, Hu X, Ren X, Shen Z, Ouyang W, Zhang Z | 2018 | Lineage tracking reveals dynamic relationships of T cells in colorectal cancer | https://www.ncbi.nlm.nih.gov/geo/query/acc.cgi?acc=GSE108989 | NCBI Gene Expression Omnibus, GSE108989 |
| Cho JW, Hong MH, Jun SH, Kim YJ, Cho BC, Lee I, Kim HR | 2020 | Genome-wide identification of differentially methylated promoters and enhancers associated with response to anti-PD-1 therapy in non-small cell lung cancer | https://www.ncbi.nlm.nih.gov/geo/query/acc.cgi?acc=GSE126044 | NCBI Gene Expression Omnibus, GSE126044 |

*Continued on next page*

*Continued*

| Author(s) | Year | Dataset title | Dataset URL | Database and Identifier |
|---|---|---|---|---|
| Liu J, Lichtenberg T, Hoadley KA, Poisson LM, Lazar AJ, Cherniack AD, Kovatich AJ, Benz CC, Levine DA, Lee AV, Omberg L, Wolf DM, Shriver CD, Thorsson V, Hu H, The Cancer Genome Atlas Research Network | 2018 | cohort: TCGA Lung Adenocarcinoma (LUAD) | https://xenabrowser.net/datapages/?cohort=TCGA%20Lung%20Adenocarcinoma%20(LUAD)&removeHub=https%3A%2F%2Fxena.treehouse.gi.ucsc.edu%3A443 | UCSC Genome Browser, TCGA-LUAD |
| Liu J, Lichtenberg T, Hoadley KA, Poisson LM, Lazar AJ, Cherniack AD, Kovatich AJ, Benz CC, Levine DA, Lee AV, Omberg L, Wolf DM, Shriver CD, Thorsson V, Hu H, The Cancer Genome Atlas Research Network | 2018 | cohort: TCGA Colon Cancer (COAD) | https://xenabrowser.net/datapages/?cohort=TCGA%20Colon%20Cancer%20(COAD)&removeHub=https%3A%2F%2Fxena.treehouse.gi.ucsc.edu%3A443 | UCSC Genome Browser, TCGA-COAD |
| Liu J, Lichtenberg T, Hoadley KA, Poisson LM, Lazar AJ, Cherniack AD, Kovatich AJ, Benz CC, Levine DA, Lee AV, Omberg L, Wolf DM, Shriver CD, Thorsson V, Hu H, The Cancer Genome Atlas Research Network | 2018 | cohort: TCGA Melanoma (SKCM) | https://xenabrowser.net/datapages/?cohort=TCGA%20Melanoma%20(SKCM)&removeHub=https%3A%2F%2Fxena.treehouse.gi.ucsc.edu%3A443 | UCSC Genome Browser, TCGA-SKCM |

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
