## [Editor Report · eLife Assessment]

MGPfact^XMBD^ is a novel computational method for investigating cell evolutionary trajectory for scRNA-seq samples. It is **important**, with several potential future applications. The authors benchmarked this method using synthetic and real-world samples and showed superior performance for some of the tasks in cell trajectory analysis compared to other methods with **compelling** evidence.

---

## [Referee Report · Reviewer #1 (Public review)]

Summary:

Ren et al developed a novel computational method to investigate cell evolutionary trajectory for scRNA-seq samples. This method, MGPfact, estimates pseudotime and potential branches in the evolutionary path through explicitly modeling the bifurcations in a Gaussian process. They benchmarked this method using synthetic as well as real world samples and showed superior performance for some of the tasks in cell trajectory analysis. They further demonstrated the utilities of MGPfact using single cell RNA-seq samples derived from microglia or T cells and showed that it can accurately identify the differentiation timepoint and uncover biologically relevant gene signatures.

Strengths:

Overall I think this is a useful new tool that could deliver novel insights for the large body of scRNA-seq data generated in the public domain. The manuscript is written is a logical way and most parts of the method are well described.

Comments on revisions:

In this revision, the authors have sufficiently addressed all of my concerns. I don't have any follow-up comments.

---

## [Referee Report · Reviewer #2 (Public review)]

Summary of the manuscript:

Authors present MGPfact^XMBD^, a novel model-based manifold-learning framework designed to address the challenges of interpreting complex cellular state spaces from single-cell RNA sequences. To overcome current limitations, MGPfactXMBD factorizes complex development trajectories into independent bifurcation processes of gene sets, enabling trajectory inference based on relevant features. As a result, it is expected that the method provides a deeper understanding of the biological processes underlying cellular trajectories and their potential determinants.

MGPfact^XMBD^ was tested across 239 datasets, and the method demonstrated similar to slightly superior performance in key quality-control metrics to state-of-the-art methods. When applied to case studies, MGPfact^XMBD^ successfully identified critical pathways and cell types in microglia development, validating experimentally identified regulons and markers. Additionally, it uncovered evolutionary trajectories of tumor-associated CD8+ T cells, revealing new subtypes with gene expression signatures that predict responses to immune checkpoint inhibitors in independent cohorts.

Overall, MGPfact^XMBD^ represents a relevant tool in manifold-learning for scRNA-seq data, enabling feature selection for specific biological processes and enhancing our understanding of the biological determinants of cell fate.

Summary of the outcome:

The novel method addresses core state-of-the-art questions in biology related to trajectory identification. The design and the case studies are of relevance.

Comments on revisions:

The authors have addressed all my previous comments to satisfaction.

---

## [Author Response]

The following is the authors’ response to the original reviews.

**Reviewer #1:**
Comment#1: Ren et al developed a novel computational method to investigate cell evolutionary trajectory for scRNA-seq samples. This method, MGPfact, estimates pseudotime and potential branches in the evolutionary path by explicitly modeling the bifurcations in a Gaussian process. They benchmarked this method using synthetic as well as real-world samples and showed superior performance for some of the tasks in cell trajectory analysis. They further demonstrated the utilities of MGPfact using single-cell RNA-seq samples derived from microglia or T cells and showed that it can accurately identify the differentiation timepoint and uncover biologically relevant gene signatures. Overall I think this is a useful new tool that could deliver novel insights for the large body of scRNA-seq data generated in the public domain. The manuscript is written in a logical way and most parts of the method are well described.

Thank you for reviewing our manuscript and for your positive feedback on MGPfact. We are pleased that you find it useful for identifying differentiation timepoints and uncovering gene signatures. We will continue to refine MGPfact and explore its applications across diverse datasets. Your insights are invaluable, and we appreciate your support.

Comment#2: Some parts of the methods are not clear. It should be outlined in detail how pseudo time T is updated in Methods. It is currently unclear either in the description or Algorithm 1.

Thanks to the reviewers' comments. We've added a description of how pseudotime *T* is obtained between lines 138 and 147 in the article. In brief, the pseudotime of MGPfact is inferred through Gaussian process regression on the downsampled single-cell transcriptomic data. Specifically, *T* is treated as a continuous variable representing the progression of cells through the differentiation process. We describe the relationship between pseudotime and expression data using the formula:yl∗=f(T)+ε

where *f*(***T***) is a Gaussian Process (GP) with covariance matrix ***S***, and ε represents the error term. The Gaussian process is defined as:f(T)=GP(0,S+σS2⋅I)

where σS2 is the variance set to 1e-6.

During inference, we update the pseudotime by maximizing the posterior likelihood. Specifically, the posterior distribution of pseudotime ***T*** can be represented as:p(T∣Y∗)∝p(Y∗∣f(T))⋅p(f(T))

where p(Y∗∣f(T)) is the likelihood function of the observed data *Y**, and p(f(T)) is the prior distribution of the Gaussian process. This posterior distribution integrates the observed data with model priors, enabling inference of pseudotime and trajectory simultaneously. Due to the high autocorrelation of *T* in the posterior distribution, we use Adaptive Metropolis within Gibbs (AMWG) sampling (Roberts and Rosenthal, 2009; Tierney, 1994). Other parameters are estimated using the more efficient SLICE sampling technique (Neal, 2003).

Comment#3: There should be a brief description in the main text of how synthetic data were generated, under what hypothesis, and specifically how bifurcation is embedded in the simulation.

Thank you for the reviewers' comments. We have added descriptions regarding the synthetic dataset in the methods section. The revised content is from line 487 to 493:

“The synthetic datasets were generated using four simulators: dyngen (Saelens et al., 2019), dyntoy (Saelens et al., 2019), PROSSTT (Papadopoulos et al., 2019), and Splatter (Zappia et al., 2017), each modeling different trajectory topologies such as linear, branching, and cyclic. Splatter simulates branching events by setting expression states and transition probabilities, dyntoy generates random expression gradients to reflect dynamic changes, and dyngen focuses on complex branching structures within gene regulatory networks.”

Comment#4: Please explain what the abbreviations mean at their first occurrence.

We appreciate the reviewers' feedback. We have thoroughly reviewed the entire manuscript and made sure that all abbreviations have had their full forms provided upon their first occurrence.

Comment#5: In the benchmark analysis (Figures 2/3), it would be helpful to include a few trajectory plots of the real-world data to visualize the results and to evaluate the accuracy.

We appreciate the reviewer's feedback. To more clearly demonstrate the performance of MGPfact, we selected three representative cases from the dataset for visual comparison. These cases represent different types of trajectory structures: linear, bifurcation, and multifurcation. The revised content is between line 220 and 226.

As shown in Supplementary Fig. 5, it is evident that MGPfact excels in capturing main developmental paths and identifying key bifurcation points. In the linear trajectory structure, MGPfact accurately predicted the linear structure without bifurcation events, showing high consistency with the ground truth (*overall*=0.871). In the bifurcation trajectory structure, MGPfact accurately captured the main bifurcation event (*overall*=0.636). In the multifurcation trajectory structure, although MGPfact predicted only one bifurcation point, its overall structure remains close to the ground truth, as evidenced by its high overall score (*overall*=0.566). Overall, MGPfact demonstrates adaptability and accuracy in reconstructing various types of trajectory structures.

Comment#6: It is not clear how this method selects important genes/features at bifurcation. This should be elaborated on in the main text.

Thanks to the reviewers' comments. To enhance understanding, we've added detailed descriptions of gene selection in the main text and appendix, specifically from lines 150 to 161. In brief, MGPfact employs a Gaussian process mixture model to infer cell fate trajectories and identify independent branching events. We calculate load matrices using formulas 1 and 14 to assess each gene's contribution to the trajectories. Genes with an absolute weight greater than 0.05 are considered predominant in specific branching processes. Subsequently, SCENIC (Aibar et al., 2017; Bravo González-Blas et al., 2023) analysis was conducted to further infer the underlying regulons and annotate the biological processes of these genes.

Comment#7: It is not clear how survival analysis was performed in Figure 5. Specifically, were critical confounders, such as age, clinical stage, and tumor purity controlled?

To evaluate the predictive and prognostic impacts of the selected genes, we utilized the Cox multivariate regression model, where the effects of relevant covariates, including age, clinical stage, and tumor purity, were adjusted. We then conducted the Kaplan-Meier survival analysis again to ensure the reliability of the results. The revisions mainly include the following sections:

(1) We modified the description of adjusting for confounding factors in the survival analysis, from line 637 to 640:

“To adjust for possible confounding effects, the relevant clinical features including age, sex and tumor stage were used as covariates. The Cox regression model was implemented using R-4.2 package “survival”. And we generated Kaplan-Meier survival curves based on different classifiers to illustrate differences in survival time and report the statistical significance based on Log-rank test.”

(2) We updated the images in the main text regarding the survival analysis, including Fig. 5a-b, Fig. 6c, and Supplementary Fig. 8e.

Comment#8: I recommend that the authors perform some sort of 'robustness' analysis for the consensus tree built from the bifurcation Gaussian process. For example, subsample 80% of the cells to see if the bifurcations are similar between each bootstrap.

We appreciate the reviewers' feedback. We performed a robustness analysis of the consensus tree using 100 training datasets. This involved sampling the original data at different proportions, and then calculating the topological similarity between the consensus trajectory predictions of MGPfact and those without sampling, using the Hamming-Ipsen-Mikhailov (*HIM*) metric. A higher score indicates greater robustness. The relevant figure is in Supplementary Fig. 4, and the description is in the main text from line 177 to 182.

The results indicate that the consensus trajectory predictions based on various sampling proportions of the original data maintain a high topological similarity with the unsampled results (*HIM_mean_*=0.686). This demonstrates MGPfact’s robustness and generalizability under different data conditions, hence the capability of capturing bifurcative processes in the cells’ trajectory.

**Reviewer #2:**
Comment#1: The authors present MGPfact^XMBD^, a novel model-based manifold-learning framework designed to address the challenges of interpreting complex cellular state spaces from single-cell RNA sequences. To overcome current limitations, MGPfact^XMBD^ factorizes complex development trajectories into independent bifurcation processes of gene sets, enabling trajectory inference based on relevant features. As a result, it is expected that the method provides a deeper understanding of the biological processes underlying cellular trajectories and their potential determinants. MGPfact^XMBD^ was tested across 239 datasets, and the method demonstrated similar to slightly superior performance in key quality-control metrics to state-of-the-art methods. When applied to case studies, MGPfact^XMBD^ successfully identified critical pathways and cell types in microglia development, validating experimentally identified regulons and markers. Additionally, it uncovered evolutionary trajectories of tumor-associated CD8+ T cells, revealing new subtypes with gene expression signatures that predict responses to immune checkpoint inhibitors in independent cohorts. Overall, MGPfact^XMBD^ represents a relevant tool in manifold learning for scRNA-seq data, enabling feature selection for specific biological processes and enhancing our understanding of the biological determinants of cell fate.

Thank you for your thoughtful review of our manuscript. We are thrilled to hear that you find MGPfact^XMBD^ beneficial for exploring cellular evolutionary paths in scRNA-seq data. Your insights are invaluable, and we look forward to incorporating them to further enrich our study. Thank you once again for your support and constructive feedback.

Comment#2: How the methods compare with existing Deep Learning based approaches such as TIGON is a question mark. If a comparison would be possible, it should be conducted; if not, it should be clarified why.

We appreciate the reviewer's comments. We have added a comparison with the sctour (Li, 2023) and TIGON methods (Sha, 2024).

It is important to note that the encapsulation and comparison of MGPfact are based on traditional differentiation trajectory construction. Saelens et al. established a systematic evaluation framework that categorizes differentiation trajectory structures into topological subtypes such as linear, bifurcation, multifurcation, graph, and tree, focusing on identifying branching structures in the cell differentiation process (Saelens et al., 2019). The sctour and TIGON methods mentioned by the reviewer are primarily used for estimating RNA velocity, focusing on continuous temporal evolution rather than explicit branching structures, and do not explicitly model branches. Therefore, we considered the predictions of these two methods as linear trajectories and compared them with MGPfact. While scTour explicitly estimates pseudotime, TIGON uses the concept of "growth," which is analogous to pseudotime, so we made the necessary adaptations.

Author response image 1 show that within this framework, compared to scTour (*overall_mean_*=0.448) and TIGON (*overall_mean_*=0.263), MGPfact still maintains a relatively high standard (*overall_mean_*=0.534). This indicates that MGPfact has a significant advantage in accurately capturing branching structures in cell differentiation, especially in applications where explicit modeling of branches is required.

**Author response image 1. sa3fig1:** Comparison of MGPfact with scTour and TIGON in trajectory inference performance across 239 test datasets. (**a**) Overall scores; (**b**) *F*1*_branches_*; (**c**) *HIM*; (**d**) *cor*_*dist*_; (**e**) *wcor_features_*. All results are color-coded based on the trajectory types, with the black line representing the mean value. The “Overall” assessment is calculated as the geometric mean of all four metrics.

Comment#3: Missing Methods:- The paper lacks a discussion of Deep Learning approaches for bifurcation analysis. e.g. scTour, Tigon.- I am missing comments on methods such CellRank, and alternative approaches to delineate a trajectory.

We thank the reviewer for these comments.

(1) As mentioned in response to Comments#2, the scTour and TIGON methods are primarily used for estimating RNA velocity, focusing on continuous temporal evolution rather than explicit branching structures, and they do not explicitly model branches. We consider the predictions of these two methods as linear trajectories and compare them with MGPfact. The relevant description and discussion have been addressed in the response.

(2) We have added a description of RNA velocity estimation methods (scTour, TIGON, CellRank) in the introduction section. The revised content is from line 66 to 71:

Moreover, recent studies based on RNA velocity has provided insights into cell state transitions. These methods measure RNA synthesis and degradation rates based on the abundance of spliced and unspliced mRNA, such as CellRank (Lange et al., 2022). Nevertheless, current RNA velocity analyses are still unable to resolve cell-fates with complex branching trajectory. Deep learning methods such as scTour (Li, 2023) and TIGON (Sha, 2024) circumvent some of these limitations, offering continuous state assumptions or requiring prior cell sampling information.

Comment#4: Impact of MURP:The rationale for using MURP is well-founded, especially for trajectory definition. However, its impact on the final results needs evaluation.How does the algorithm compare with a random subselection of cells or the entire cell set?

Thank you for the comments. We fully agree that MURP is crucial in trajectory prediction. As a downsampling method, MURP is specifically designed to address noise issues in single-cell data by dividing the data into several subsets, thereby maximizing noise reduction while preserving the main structure of biological variation (Ren et al., 2022). In MGPfact, MURP typically reduces the data to fewer than 100 downsampled points, preserving the core biological structure while lowering computational complexity. To assess MURP's impact, we conducted experiments by randomly selecting 20, 40, 60, 80, and 100 cells for trajectory inference. These results were mapped back to the original data using the KNN graph structure for final predictions, which were then compared with the MURP downsampling results. Supplementary results can be found in Supplementary Fig. 3, with additional descriptions in the main text from line 170 to 176.

The results indicate that trajectory inference using randomly sampled cells has significantly lower prediction accuracy compared to that using MURP. This is particularly evident in branch assignment (*F*1*_branches_*) and correlation *cor_dist_*, where the average levels decrease by 20.5%-64.9%. In contrast, trajectory predictions using MURP for downsampling show an overall score improvement of 5.31%-185%, further highlighting MURP's role in enhancing trajectory inference within MGPfact.

Comment#5: What is the impact of the number of components selected?

Thank you for the comments. In essence, MGPfact consists of two main steps: (1) trajectory inference; (2) calculation of factorized scores and identification of high-weight genes. After step 1, MGPfact estimates parameters such as pseudotime ***T*** and bifurcation points ***B***. In step 2, we introduce a rotation matrix R={r1,r2,…,rL} to obtain factor scores for each trajectory *l* by rotating ***Y****.wl=Y∗⋅rl+el2

For all trajectories,p(W∣Y∗)=∏l=1L[N(Y∗⋅rl+el∣0,sl)⋅N(el∣0,σerror 2)]

where el is the error term for the l-th trajectory. The number of features in ***Y**** must match the dimensions of the rotation matrix ***R*** to ensure the factorized score matrix ***W*** contains factor scores for *L* trajectories, achieving effective feature representation and interpretation in the model.

Additionally, to further illustrate the impact of the number of principal components (PCs) on model performance in step 1, we conducted additional experiments. We used 3 PCs as the default and adjusted the number to evaluate changes from this baseline. As shown in Author response image 2, setting the number of PCs to 1 significantly decreases the overall performance score (*Overall_mean_*=0.363), as well as the *wcor_features_* (wcorfeaturesmean=0.586) and *cor_dist_* (cordistmean=0.21) metrics. In contrast, increasing the number of PCs does not significantly affect the metrics. It ought to be mentioned that number of components used should be determined by the intrinsic biological characteristics of the cell fate-determination. Our experiment based on a limited number of datasets may not represent more complex scenarios in other cell types.

**Author response image 2. sa3fig2:** Robustness testing of the number of MURP PCA components on 100 training datasets. With the number of principal components (PCs) set to 3 by default; we tested the impact of different number of components (1-10) on the prediction results. In all box plots, the asterisk represents the mean value, while the whiskers extend to the farthest data points within 1.5 times the interquartile range. Significance is denoted as follows: not annotated indicates non-significant; * P < 0.05; ** P < 0.01; *** P < 0.001; two-sided paired Student’s T-tests.

Comment#6: Please comment on the selection of the kernel functions (rbf and polynomial) and explain why other options were discarded.

Thank you for the comments. We have added a description regarding the selection of radial basis functions and polynomial kernels in lines 126-130. As the reviewers mentioned, the choice of kernel functions is crucial in the MGPfact analysis pipeline for constructing the covariance matrix of the Gaussian process. We selected the radial basis function (RBF) kernel and the polynomial kernel to balance capturing data complexity and computational efficiency. The RBF kernel is chosen for its ability to effectively model smooth functions and capture local variations in the data, making it well-suited to the continuous and smooth characteristics of biological processes; its hyperparameters offer modeling flexibility. The polynomial kernel is used to capture more complex nonlinear relationships between input features, with its hyperparameters also allowing further customization of the model. In contrast, other complex kernels, such as Matérn or spectral kernels, were omitted due to their interpretability challenges and the risk of overfitting with limited data. However, as suggested by the reviewers, we will consider and test the impact of other kernel functions on the covariance matrix of the Gaussian process and their role in trajectory inference in our subsequent phases of algorithm design.

Comment#7: What is the impact of the Pseudotime method used initially? This section should be expanded with clear details on the techniques and parameters used in each analysis.

We are sorry for the confusion. We've added a description of how pseudotime is obtained between line 138 and 147 in the main text. And the specific hyperparameters involved in the model and their prior settings are detailed in the supplementary information.

In brief, the pseudotime and related topological parameters of the bifurcative trajectories in MGPfact are inferred by Gaussian process regression from downsampled single-cell transcriptomic data (MURP). Specifically, is treated as a continuous variable representing the progression of cells through the differentiation process. We describe the relationship between pseudotime and expression data as:yl∗=f(T)+ε

where f(T) is a Gaussian Process (GP) with covariance matrix S, and ε represents the error term. The Gaussian process is defined as:f(T)=GP(0,S+σS2⋅I)

where σS2 is the variance set to 1e-6. During inference, we update the pseudotime by maximizing the posterior liklihood. Specifically, the posterior distribution of pseudotime is obtained by combining the observed data ***Y**** with the prior distribution of the Gaussian process model.p(T∣Y∗)∝p(Y∗∣f(T))⋅p(f(T))

We use the Markov Chain Monte Carlo method for parameter estimation, particularly employing the adaptive Metropolis-within-Gibbs (AMWG) sampling to handle the high autocorrelation of pseudotime.

Comment#8: Enhancing Readability: For clarity, provide intuitive descriptions of each evaluation function used in simulated and real data. The novel methodology performs well for some metrics but less so for others. A clear understanding of these measurements is essential.

To address the concern of readability, we have added descriptions of 5 evaluation metrics in the methodology section (Benchmarking MGPfact to state-of-the-art methods) in line 494 to 515. Additionally, we have included a summary and discussion of these metrics in the conclusion section in line 214-240 to help the readers better understand the significance and impact of these measurements.

(1) In brief, the Hamming-Ipsen-Mikhailov (*HIM*) distance measures the similarity between topological structures, combining the normalized Hamming distance and the Ipsen-Mikhailov distance, which focus on edge length differences and degree distribution similarity, respectively. The *F1_branches_* is used to assess the accuracy of a model's branch assignment via Jaccard similarity between branch pairs. In trajectory inference, *cor_dist_* quantifies the similarity of inter-cell distances between predicted and true trajectories, evaluating the accuracy of cell ordering. The *wcor_features_* assesses the similarity of key features through weighted Pearson correlation, capturing biological variation. The *Overall* score is calculated as the geometric mean of these metrics, providing an assessment of overall performance.

(2) For MGPfact and the other seven methods included in the comparison, each has its own focus. MGPfact specializes in factorizing complex cell trajectories using Gaussian process mixture models, making it particularly capable of identifying bifurcation events. Therefore, it excels in the accuracy of branch partitioning and similarity of trajectory topology. Among other methods, scShaper (Smolander et al., 2022) and TSCAN(Ji and Ji, 2016) are more suited for generating linear trajectories and excel in linear datasets, accurately predicting pseudotime. The Monocle series, as typical representatives of tree methods, effectively capture complex topologies and are suitable for analyzing cell data with diversified differentiation paths.

Comment#9: Microglia Analysis:In Figures 3A-C, the genes mentioned in the text for each bifurcation do not always match those shown in the panels. Please confirm this.

Thank you for pointing this out. We have carefully reviewed the article and corrected the error where the genes shown in the figures did not correspond to the descriptions in the article. The specific corrections have been made between line 257 and 264:

“The first bifurcation determines the differentiated cell fates of PAM and HM, which involves a set of notable marker genes of both cell types, such as Apoe, Selplg (HM), and Gpnmb (PAM). The second bifurcation determines the proliferative status, which is crucial for the development and function of PAM and HM (Guzmán, n.d.; Li et al., 2019). The genes affected by the second bifurcation are associated with cell cycle and proliferation, such as Mki67, Tubb5, Top2a. The third bifurcation influences the development and maturity of microglia, of which the highly weighted genes, such as Tmem119, P2ry12, and Sepp1 are all previously annotated markers for establishment of the fates of microglia (Anderson et al., 2022; Li et al., 2019) (Supplementary Table 4).”

Comment#10: Regulons:- The conclusions rely heavily on regulons. The Methods section describes using SCENIC, GENIE3, RCisTarget, and AUCell, but their relation to bifurcation analysis is unclear.- Do you perform trajectory analysis on all MURP-derived cells or within each identified trajectory based on bifurcation? This point needs clarification to make the outcomes comprehensible. The legend of Figure 4 provides some ideas, but further clarity is required.

Thank you for the comments.

(1) To clarify, we used the tools like SCENIC to annotate the highly weighted genes (HWG) resulted from the bifurcation analysis for transcription factor regulation activity and possible impacts on biological processes. We have added descriptions to the analysis of our microglial data. The revised content is between line 265 and 266:

Moreover, we retrieved highly active regulons from the HWG by MGPfact, of which the significance is quantified by the overall weights of the member genes.

(2) We apologize for any confusion caused by our description. It is important to clarify that we performed an overall trajectory analysis on all MURP results, rather than analyzing within each identified trajectory. Specifically, we first used MURP to downsample all preprocessed cells, where each MURP subset represents a group of cells. We then conducted trajectory inference on all MURP subsets and identified bifurcation points. This process generated multiple independent differentiation trajectories, encompassing all MURP subsets. To clearly convey this point, we have added descriptions in the legend of Figure 4. The revised content is between line 276 and 283:

“Fig. 4. MGPfact reconstructed the developmental trajectory of microglia, recovering known determinants of microglia fate. a-c. The inferred independent bifurcation processes with respect to the unique cell types (color-coded) of microglia development, where phase 0 corresponds to the state before bifurcation; and phases 1 and 2 correspond to the states post-bifurcation. Each colored dot represents a metacell of unique cell type defined by MURP. The most highly weighted regulons in each trajectory were labeled by the corresponding transcription factors (left panels). The HWG of each bifurcation process include a set of highly weighted genes (HWG), of which the expression levels differ significantly among phases 1, 2, and 3 (right panels).”

Comment#11: CD8+ T Cells: The comparison is made against Monocle2, the method used in the publication, but it would be beneficial to compare it with more recent methods. Otherwise, the added value of MGPfact is unclear.

Per your request, we have expanded our comparative analysis to include not only Monocle2 but also more recent methods such as Monocle3 (Cao et al., 2019) and scFates Tree (Faure et al., 2023). We used adjusted R-squared values to evaluate each method's ability to explain trajectory variation. The results have been added to Table 2 and Supplementary Table 6. The revised content is between line 318 and 326:

“We assessed the goodness-of-fit (adjusted R-square) of the consensus trajectory derived by MGPfact and three methods (Monocle 2, Monocle 3 and scFates Tree) for the CD8+ T cell subtypes described in the original studies (Guo et al., 2018; Zhang et al., 2018). The data showed that MGPfact significantly improved the explanatory power for most CD8+ T cell subtypes over Monocle 2, which was used in the original studies (P < 0.05, see Table 2 and Supplementary Table 6), except for the CD8-GZMK cells in the CRC dataset. Additionally, MGPfact demonstrated better explanatory power in specific cell types when compared to Monocle 3 and scFates Tree. For instance, in the NSCLC dataset, MGPfact exhibited higher explanatory power for CD8-LEF1 cells (Table 2, R-squared = 0.935), while Monocle 3 and scFates Tree perform better in other cell types.”

Comment#12: Consensus Trajectory: A panel explaining how the consensus trajectory is generated would be helpful. Include both visual and textual explanations tailored to the journal's audience.

Thank you for the comments. Regarding how the consensus trajectory is constructed, we have illustrated and described this in Figure 1 and the supplementary methods. Taking the reviewers' suggestions into account, we have added more details about the generation process of the consensus trajectory in the methods section to enhance the completeness of the manuscript. The revised content is from line 599 to 606:

“Following MGPfact decomposition, we obtained multiple independent bifurcative trajectories, each corresponds to a binary tree within the temporal domain. These trajectories were then merged to construct a coherent diffusion tree, representing the consensus trajectory of cells’ fate. The merging process involves initially sorting all trajectories by their bifurcation time. The first (earliest) bifurcative trajectory is chosen as the initial framework, and subsequent trajectories are integrated to the initial framework iteratively by adding the corresponding branches at the bifurcation timepoints. As a result, the trajectories are ultimately merged into a comprehensive binary tree, serving as the consensus trajectory.”

Comment#13: Discussion:- Check for typos, e.g., line 382 "pseudtime.".- Avoid considering HVG as the entire feature space.- The first three paragraphs are too similar to the Introduction. Consider shortening them to succinctly state the scenario and the implications of your contribution.

Thank you for pointing out the typos.

(1) We conducted a comprehensive review of the document to ensure there are no typographical errors.

(2) We restructured the first three paragraphs of the discussion section to clarify the limitations in the use of current manifold-learning methods and removed any absolute language regarding treating HVGs as the entire feature space. The revised content is from line 419 to 430:

“Single-cell RNA sequencing (scRNA-seq) provides a direct, quantitative snapshot of a population of cells in certain biological conditions, thereby revealing the actual cell states and functions. Although existing clustering and embedding algorithms can effectively reveal discrete biological states of cells, these methods become less efficient when depicting continuous evolving of cells over the temporal domain. The introduction of manifold learning offers a new dimension for discovery of relevant biological knowledge in cell fate determination, allowing for a better representation of continuous changes in cells, especially in time-dependent processes such as development, differentiation, and clonal evolution. However, current manifold learning methods face major limitations, such as the need for prior information on pseudotime and cell clustering, and lack of explainability, which restricts their applicability. Additionally, many existing trajectory inference methods do not support gene selection, making it difficult to annotate the results to known biological entities, thereby hindering the interpretation of results and subsequent functional studies.”

Comment#14: Minor Comments:(1) Review the paragraph regarding the "current manifold-learning methods are faced with two major challenges." The message needs clarification.(2) Increase the quality of the figures.(3) Update the numbering of equations from #(.x) to (x).

We thank the reviewer for these detailed suggestions.

(1) We have thoroughly revised the discussion section, addressing overly absolute statements. The revised content is from line 426 to 428:

“However, current manifold learning methods face major limitations, such as the need for prior information on pseudotime and cell clustering, and lack of explainability, which restricts their applicability.”

(2) We conducted a comprehensive review of the figures in the article to more clearly present our results.

(3) We have meticulously reviewed the equations in the article to ensure there are no display issues with the indices.

References

Aibar S, González-Blas CB, Moerman T, Huynh-Thu VA, Imrichova H, Hulselmans G, Rambow F, Marine J-C, Geurts P, Aerts J, van den Oord J, Atak ZK, Wouters J, Aerts S. 2017. SCENIC: single-cell regulatory network inference and clustering. *Nat Methods* 14:1083–1086. doi:10.1038/nmeth.4463

Anderson SR, Roberts JM, Ghena N, Irvin EA, Schwakopf J, Cooperstein IB, Bosco A, Vetter ML. 2022. Neuronal apoptosis drives remodeling states of microglia and shifts in survival pathway dependence. *Elife* 11:e76564.

Bravo González-Blas C, De Winter S, Hulselmans G, Hecker N, Matetovici I, Christiaens V, Poovathingal S, Wouters J, Aibar S, Aerts S. 2023. SCENIC+: single-cell multiomic inference of enhancers and gene regulatory networks. *Nat Methods*. doi:10.1038/s41592-023-01938-4

Cao J, Spielmann M, Qiu X, Huang X, Ibrahim DM, Hill AJ, Zhang F, Mundlos S, Christiansen L, Steemers FJ, Trapnell C, Shendure J. 2019. The single-cell transcriptional landscape of mammalian organogenesis. *Nature* 566:496–502. doi:10.1038/s41586-019-0969-x

Faure L, Soldatov R, Kharchenko PV, Adameyko I. 2023. scFates: a scalable python package for advanced pseudotime and bifurcation analysis from single-cell data. *Bioinformatics* 39:btac746. doi:10.1093/bioinformatics/btac746

Guo X, Zhang Y, Zheng L, Zheng C, Song J, Zhang Q, Kang B, Liu Z, Jin L, Xing R, Gao R, Zhang L, Dong M, Hu X, Ren X, Kirchhoff D, Roider HG, Yan T, Zhang Z. 2018. Global characterization of T cells in non-small-cell lung cancer by single-cell sequencing. *Nat Med* 24:978–985. doi:10.1038/s41591-018-0045-3

Guzmán AU. n.d. Single-cell RNA sequencing of spinal cord microglia in a mouse model of neuropathic pain.

Ji Z, Ji H. 2016. TSCAN: Pseudo-time reconstruction and evaluation in single-cell RNA-seq analysis. *Nucleic Acids Res* 44:e117–e117. doi:10.1093/nar/gkw430

Lange M, Bergen V, Klein M, Setty M, Reuter B, Bakhti M, Lickert H, Ansari M, Schniering J, Schiller HB, Pe’er D, Theis FJ. 2022. CellRank for directed single-cell fate mapping. *Nat Methods* 19:159–170. doi:10.1038/s41592-021-01346-6

Li Q. 2023. scTour: a deep learning architecture for robust inference and accurate prediction of cellular dynamics. *Genome Biology*.

Li Q, Cheng Z, Zhou L, Darmanis S, Neff NF, Okamoto J, Gulati G, Bennett ML, Sun LO, Clarke LE, Marschallinger J, Yu G, Quake SR, Wyss-Coray T, Barres BA. 2019. Developmental Heterogeneity of Microglia and Brain Myeloid Cells Revealed by Deep Single-Cell RNA Sequencing. *Neuron* 101:207-223.e10. doi:10.1016/j.neuron.2018.12.006

Neal RM. 2003. Slice sampling. *The annals of statistics* 31:705–767.

Papadopoulos N, Gonzalo PR, Söding J. 2019. PROSSTT: probabilistic simulation of single-cell RNA-seq data for complex differentiation processes. *Bioinformatics* 35:3517–3519. doi:10.1093/bioinformatics/btz078

Ren J, Zhang Q, Zhou Y, Hu Y, Lyu X, Fang H, Yang J, Yu R, Shi X, Li Q. 2022. A downsampling method enables robust clustering and integration of single-cell transcriptome data. *Journal of Biomedical Informatics* 130:104093. doi:10.1016/j.jbi.2022.104093

Roberts GO, Rosenthal JS. 2009. Examples of adaptive MCMC. *Journal of computational and graphical statistics* 18:349–367.

Saelens W, Cannoodt R, Todorov H, Saeys Y. 2019. A comparison of single-cell trajectory inference methods. *Nat Biotechnol* 37:547–554. doi:10.1038/s41587-019-0071-9

Sha Y. 2024. Reconstructing growth and dynamic trajectories from single-cell transcriptomics data 6.

Smolander J, Junttila S, Venäläinen MS, Elo LL. 2022. scShaper: an ensemble method for fast and accurate linear trajectory inference from single-cell RNA-seq data. *Bioinformatics* 38:1328–1335. doi:10.1093/bioinformatics/btab831

Tierney L. 1994. Markov chains for exploring posterior distributions. *the Annals of Statistics* 1701–1728.

Zappia L, Phipson B, Oshlack A. 2017. Splatter: simulation of single-cell RNA sequencing data. *Genome Biol* 18:174. doi:10.1186/s13059-017-1305-0

Zhang L, Yu X, Zheng L, Zhang Y, Li Y, Fang Q, Gao R, Kang B, Zhang Q, Huang JY, Konno H, Guo X, Ye Y, Gao S, Wang S, Hu X, Ren X, Shen Z, Ouyang W, Zhang Z. 2018. Lineage tracking reveals dynamic relationships of T cells in colorectal cancer. *Nature* 564:268–272. doi:10.1038/s41586-018-0694-x